# Comparative analysis of gene importance in *Escherichia coli* across growth conditions

Antoine Champie,[1] Simon Jeanneau,[1] Amélie De Grandmaison,[1] Mathias Martin Silva,[1] Jean-Philippe Coté,[1] Pierre-Étienne Jacques,[1,2,3] Sébastien Rodrigue[1,2,3]

**ABSTRACT** As the ability to synthesize complete genomes becomes increasingly possible, the question of what should compose those sequences is becoming more prevalent. However, identifying genes essential for the survival of an organism is challenging, as gene essentiality is a nuanced concept that heavily depends on context. In this study, we identified growth medium-specific fitness-impacting genes by performing transposon mutagenesis in *Escherichia coli* BW25113 and sequencing mutant populations at multiple time points in three growth media. Our analysis revealed a core set of 412 core genes with high impact on the fitness across all conditions, along with distinct medium-specific gene sets. By analyzing temporal variations in read counts per gene, we identified additional sets of genes whose inactivation causes an appreciable, albeit lower, impact on fitness. We used these sets to define medium-specific gene modules required to sustain robust growth under each condition. Our study underscores the context-dependent nature of gene essentiality and represents a step toward refining the concept from a universal list to a more nuanced, condition-specific framework, which will be invaluable for future genome design efforts.

**IMPORTANCE** As complete genome synthesis becomes more accessible, determining which genes should be included in a synthetic genome to provide robust growth becomes increasingly critical. This study demonstrates that gene contributions to fitness are not binary and depend strongly on environmental conditions. By analyzing *E. coli* transposon mutants grown in different media over multiple passages using an innovative sliding-window approach, we identified genes generally important for fitness and others that are condition-specific or have a reduced, yet measurable, impact on fitness. Using these gene sets, we formulated medium-specific gene modules that combine essential and fitness-contributing genes to support robust growth in each environment. This improved understanding takes us beyond static gene lists and toward dynamic, context-aware genome design tailored for specific applications.

**KEYWORDS** gene essentiality, transposon mutagenesis, *Escherichia coli*, growth conditions

As methods for genome synthesis and assembly improve (1–3), it is becoming more important to consider what sequences should be included in a synthetic genome (4). While general knowledge of biological mechanisms steadily increases over time, our understanding is still far from exhaustive. For example, despite decades of research, the role of approximately 35% of *Escherichia coli* genes remains elusive (5, 6), and their importance for robust growth is undetermined in many cases. This knowledge gap is particularly problematic for genome minimization, which involves removing non-essential elements from a genome while preserving its viability and functionality, since a single error in genome composition can compromise the viability of the entire organism. Even

Address correspondence to Sébastien Rodrigue, Sebastien.Rodrigue@USherbrooke.ca, or Pierre-Étienne Jacques, Pierre-Etienne.Jacques@USherbrooke.ca.

Antoine Champie, Simon Jeanneau, and Amelie De Grandmaison contributed equally to this article. Author order was determined both alphabetically and in order of increasing seniority.

The authors declare no conflict of interest.

See the funding table on p. 17.

a decade after the synthesis of the first minimal genome (7, 8), which was based on the naturally small chromosome of *Mycoplasma mycoides*, research is still ongoing to fully elucidate the function of all its genes (9). Replicating this achievement in more complex organisms remains a difficult task.

While a relatively small group of genes encoding key functions, such as DNA replication and gene expression, is likely to be strictly essential under any circumstances, the importance of many genes depends on environmental conditions. Metabolic genes, for example, are known to have a varying contribution to fitness depending on the composition of the growth medium. This variability underscores the importance of understanding how environmental factors influence gene essentiality, as such insights will be crucial for tailoring novel genomes for specific media and conditions.

Extensive data are already available to evaluate gene essentiality in well-studied microorganisms. In *E. coli*, for instance, different methods such as comparative genomics (10), genome-scale models (GEMs) (11), targeted gene deletions (12), transposon insertion sequencing (TIS) (13–17), as well as CRISPR interference (CRISPRi) (18) have been employed. However, significant challenges exist in integrating and comparing these data sets to accurately quantify the conditional importance of a given gene in this bacterium. One of the primary barriers to accurately comparing different data sets is the wide variation in growth conditions used to selectively grow gene-inactivated mutants. These differences encompass various factors, including medium composition, clonal vs bulk population cultures, temperature, aeration level, and the growth phases included in the experiments. Another barrier is the various strategies and thresholds employed to assign an essentiality status to a gene (19–23). Finally, the specific strains used to generate gene essentiality data sets can also vary, which may also cause discrepancies due to differences in genetic backgrounds (18, 24). Ultimately, even if the data available on gene essentiality are particularly extensive in *E. coli*, pinpointing the causes behind observed variations between studies remains a complex task.

In an attempt to reliably measure and compare the impact of gene inactivation under different growth conditions, we performed transposon mutagenesis in three different media using *E. coli* BW25113, the parental strain of the Keio collection of single-gene deletion mutants (12). The "MOPS minimal + 0.2% glucose" and "EZ-Rich + 0.2% glucose" defined media, respectively, abbreviated MOPS and EZR, were selected as they were specifically developed for *Enterobacteriaceae* growth (25). MOPS minimal medium is a growth medium buffered using morpholino propane sulfonate (MOPS) and Tris in which only glucose, ammonium, and base levels of metallic ions are provided. EZR medium is a MOPS medium in which all nucleic acids, amino acids, and some vitamins are also provided in defined concentrations. To identify genes required for metabolism beyond glycolysis, the primary driver of the Krebs cycle in the first two media, we also included LB (Miller) medium. LB is a commonly used, chemically undefined medium in which metabolism relies primarily on proteolysis (26). In this context, a variety of genes beyond those involved in glycolysis are required to catabolize amino acids into precursors for central metabolism. In addition, several gene essentiality data sets already exist in LB, notably those derived from the Keio collection (12) and recent TIS experiments (14), thus facilitating comparison with our results. By leveraging these diverse nutritional landscapes, this study identifies growth medium-specific fitness-impacting genes and defines the genic requirements to sustain robust growth under each condition.

## RESULTS

### Transposon mutagenesis in EZR, MOPS, and LB media

We applied the recently developed high-throughput transposon mutagenesis (HTTM) (15) method to generate 60 independent mutagenesis replicates in *E. coli* BW25113. Briefly, this protocol utilizes bacterial conjugation to deliver a transposon-carrying plasmid (pFG051) from a donor strain to the target population, establishing the initial mutant population. We generated 60 initial mutant populations (P0) which were then divided into 3 groups of 20 replicates for growth in EZR, MOPS, or LB for 5 consecutive

passages, each lasting 24 h, and referred to as P1 to P5 (Fig. 1A). As observed in previous genome streamlining endeavors, a genome composed solely of essential genes is likely to exhibit significant fitness defects and limited applicability in biotechnology (4, 27). This highlights the need to conserve additional genes that contribute to maintaining a robust organism, commonly referred to as fitness genes or quasi-essential genes (27, 28). To ensure the detection of all genes important for bacterial life (both strictly essential genes and major fitness contributors), the mutants were left to compete against the rest of the population, resulting in progressive depletion of the mutants with suboptimal fitness. The 24-h interval between each passage was intentionally designed to allow the populations to enter stationary phase in all three media, thereby eliminating mutants that could not efficiently transition between different growth phases. Because this approach identifies all genes whose inactivation affects the relative fitness of the strain across different growth phases due to various factors, we categorized them under the broader term "fitness impact" rather than using the traditional "essential genes" terminology.

To observe the gradual depletion of transposons in genes with a low fitness impact over time, as recently observed in *Acinetobacter baylyi* (29), the remaining cells were harvested after each passage (P0–P5), frozen, and their DNA extracted. Samples were then prepared for next-generation sequencing, and the transposon insertion sites were

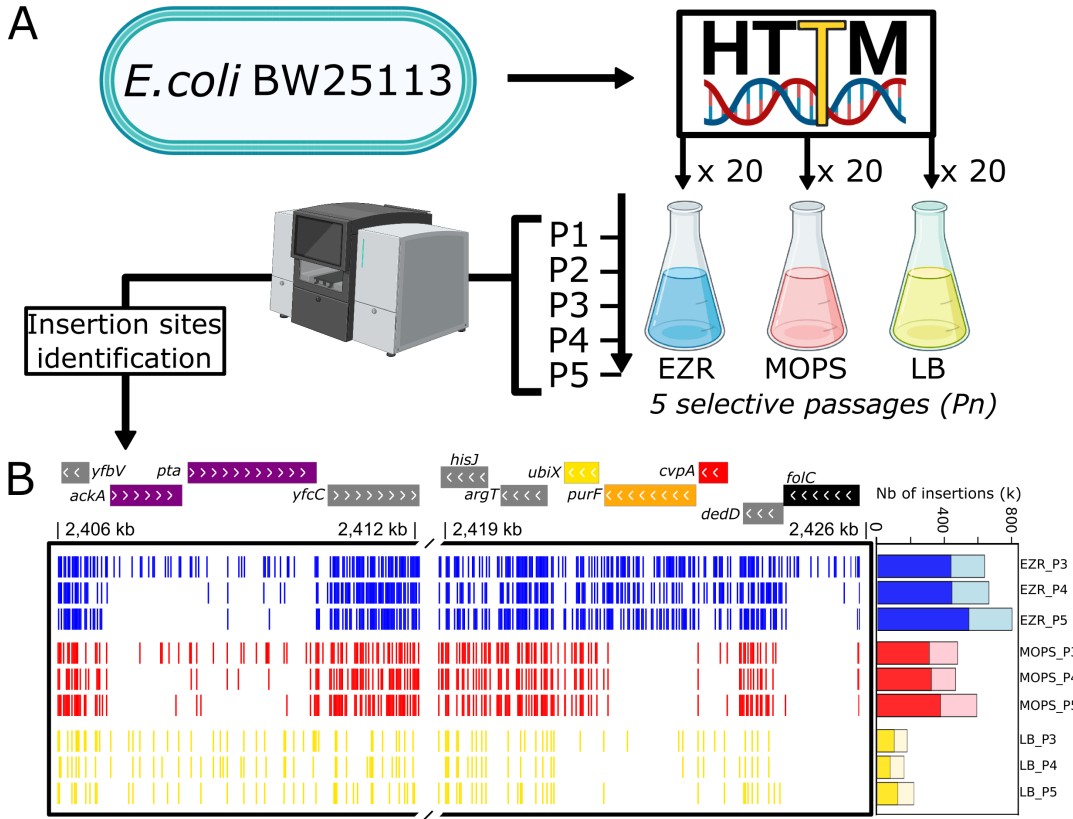

**FIG 1** Overview of the experimental strategy used in this study. (A) Schematic description of the transposon mutagenesis experimental procedure using the HTTM method. A total of five culture passages were performed for each medium (P1 to P5). (B) Representative genomic loci displaying mapped transposon insertion sites (vertical bars) across passages three to five in each tested culture medium. Genes with different transposon insertion densities such as *yfcC* (densely riddled by transposon in all three media), *pta* (showing a gradual reduction of insertions over passages), and *folC* (considered essential in the Keio collection and devoid of transposons in all three conditions at P5) can be regrouped according to the medium or media in which they have been determined to have an impact on fitness (gray = no fitness impact, black = high fitness impact important in all media, red = high fitness impact in MOPS only, yellow = high fitness impact important in LB only, orange = high fitness impact important in MOPS and LB, purple = high fitness impact in MOPS and EZR). The total number of insertions detected per passage among all pooled replicates is displayed on the right. The pale colors indicate total insertions while solid colors correspond to genic insertions. The word "number" is abbreviated as "Nb."

identified as previously described (15). For each medium, all 20 replicates of the final passages and 12 replicates of each intermediate passage (P0–P4) were sequenced, totaling 180 sequenced samples.

Early passages (P0–P2) were dominated by reads from the pFG051 mutagenesis plasmid (>95%), resulting in low genomic insertion counts (Table S1). This predominance is caused by an abundance of intact plasmid carried in multiple copies by the donor cells which have not yet been diluted enough by passages. To mitigate this phenomenon, we implemented an additional depletion step to the previously described library preparation process (15) (see Materials and Methods for additional details). Briefly, we used the AvrII restriction enzyme to cleave the intact plasmids while leaving any transposon inserted into the genome intact as there are only 16 AvrII sites in the *E. coli* genome. This additional step increased the proportion of genomic reads from barely detectable (<5% at passage 1) to an acceptable level (>30%) (Table S1).

Reads from all replicates for each condition were then pooled and used to identify transposon genomic insertion sites. This yielded a considerable number of insertions (at P5; EZR = 837,713; MOPS = 602,505; and LB = 229,591) (Fig. 1B; Fig. S1; Table S1). Considering the genome size of *E. coli* BW25113, this corresponds to an average of one insertion every 5.5 bp, 7.7 bp, and 20.0 bp in EZR, MOPS, and LB medium, respectively. The location of those insertions can be visualized through the provided UCSC genome browser public track hub (https://genome.ucsc.edu/cgi-bin/hgHubConnect?hgHub_do_redirect=on&hgHubConnect.remakeTrackHub=on&hgHub_do_firstDb=1&hubUrl=https://g-f2b62d.6d81c.5898.data.globus.org/Champie_2025/Champie_2025.hub.txt).

## Definition of a fitness impact metric

Many different methods have been described for analyzing transposon mutagenesis data and assigning an essentiality status to genes (20–22, 30, 31). These approaches usually rely on the ability to discriminate between gene populations with high vs low numbers of transposon insertions using various statistical tools. Different metrics can be used to quantify the number of genic insertions, and simple ones such as reads per gene and insertions per gene are convenient but not particularly robust, as the former is particularly sensitive to sparse insertions supported by abnormally high number of reads, while the latter is more affected by background noise insertions characterized by very low read counts.

Given the high transposon insertion density and sequencing depth of our data sets, we reasoned that alternative strategies leveraging both read counts and insertion site numbers could be explored to give a more nuanced estimation of each gene's impact on fitness, while mitigating the aforementioned pitfalls. Therefore, we developed a new method that considers both read and insertion site counts to quantify the proportion of each gene harboring no or only background levels of insertions, which is then used as a proxy to estimate the importance of each gene for cell fitness. Briefly, all genes are scanned using overlapping sliding windows (bins) whose parameters are adjusted for each sample. The bin size is determined using the average distance between transposon insertion sites, and the threshold is calculated based on the median number of reads per insertion (Materials and Methods). Each bin is then evaluated and classified either as HIT (highly inserted segment) or MISS (minimally inserted segment) depending on whether the total number of reads in the bin is above or below the defined threshold (Fig. 2A and B; Fig. S2). A fitness impact score (fiScore) is then calculated based on the proportion of MISS bins in a gene, where a score of 100% represents a gene that does not tolerate any significant level of transposon insertions anywhere on its sequence. The distribution of the number of genes per fiScore per gene is initially nearly uniform across all experimental conditions (where fiScore > 0) and transitions into a clear bimodal distribution as the number of passages increases (Fig. S3). The left mode of this final distribution consists of genes having little to no contribution to the fitness and can be modeled using an exponentially decreasing distribution (Fig. S4). For each medium, genes were defined as

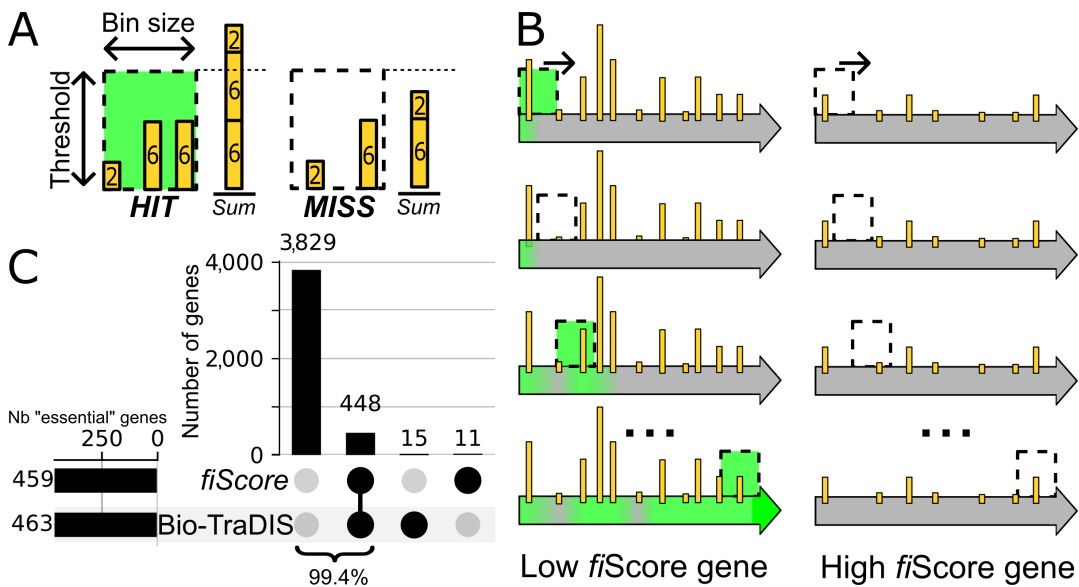

**FIG 2** Bin processing of transposon insertion sites. (A) Details on data-driven determination of bin status. A bin is labeled as "HIT" if the number of reads exceeds the threshold and otherwise labeled as "MISS." The threshold and bin size for each condition are available in Fig. S3. Yellow bars represent single base pair-resolution transposon insertion sites, and the height of the bars represents the number of mapped reads at a given position. (B) Graphical representation of the gene scanning procedure and fiScore attribution. (C) Upset plot comparing the number of genes categorized as Hi-FI in EZR after passage 5 with data processed using the fiScore or categorized as essential through the Bio-TraDIS (21) methodology. Each column represents a group of genes, with black circles indicating the method(s) that identified genes in this group as essential. The word "number" is abbreviated as "Nb."

having a "high fitness impact" (Hi-FI) if they had a probability <0.01% of belonging to the low fitness impact population according to the fitted distribution (Fig. S4). The complete fiScore data set for the last passage in all media is available in Table S2. Genes classified as Hi-FI in all three media form the core gene set.

The comparison of this new approach to the well-established Bio-TraDIS tool (21) has shown a very high agreement rate for EZR and MOPS media (99.4% and 98.3%, respectively), while the concordance in LB medium was lower (91.3%) but still very high (Fig. 2C; Fig. S5). To further support our classification, we compared Hi-FI gene lists to metabolic modeling predictions. For EZR (Fig. 3A) and MOPS (Fig. 3B) media, we used the *i*ML1515 metabolic model, simulating each medium independently for comparison (32). However, since LB is a complex and undefined medium whose exact composition is at best roughly estimated, metabolic simulation in this context would be prone to inaccuracies. The results for LB were, therefore, compared with existing high-quality essentiality data sets generated using this medium (12, 14) (Fig. 3C). We observed a strong agreement in all three cases, with a general trend of detecting almost all previously known genes impacting fitness (~90%), as well as 339 additional Hi-FI genes. This trend was expected considering the high sensitivity of our approach to reduction in fitness. The most enriched GO terms (Panther 19.0) in this LB-specific group are "ATP biosynthetic process" and "aerobic respiration." For example, the genes composing the *nuo* operon encode the subunits of NADH:quinone oxidoreductase (also known as respiratory complex I), a key enzyme in aerobic respiration. All genes of the *nuo* operon were previously deleted in the Keio collection and successfully cultivated in LB (12) but are found to be of high fitness impact only in LB medium in our data set (Fig. 3D). Competition assays in the selective conditions used for our HTTM experiment confirmed that the Δ*nuoF* mutant is quickly outcompeted by the parental strain in LB medium only (Fig. 3E). As this complex is central to aerobic respiration, we would have expected a similar fitness impairment across all three tested media. The reason behind this medium-specific fitness loss remains unclear, but it underscores the limitations of our current

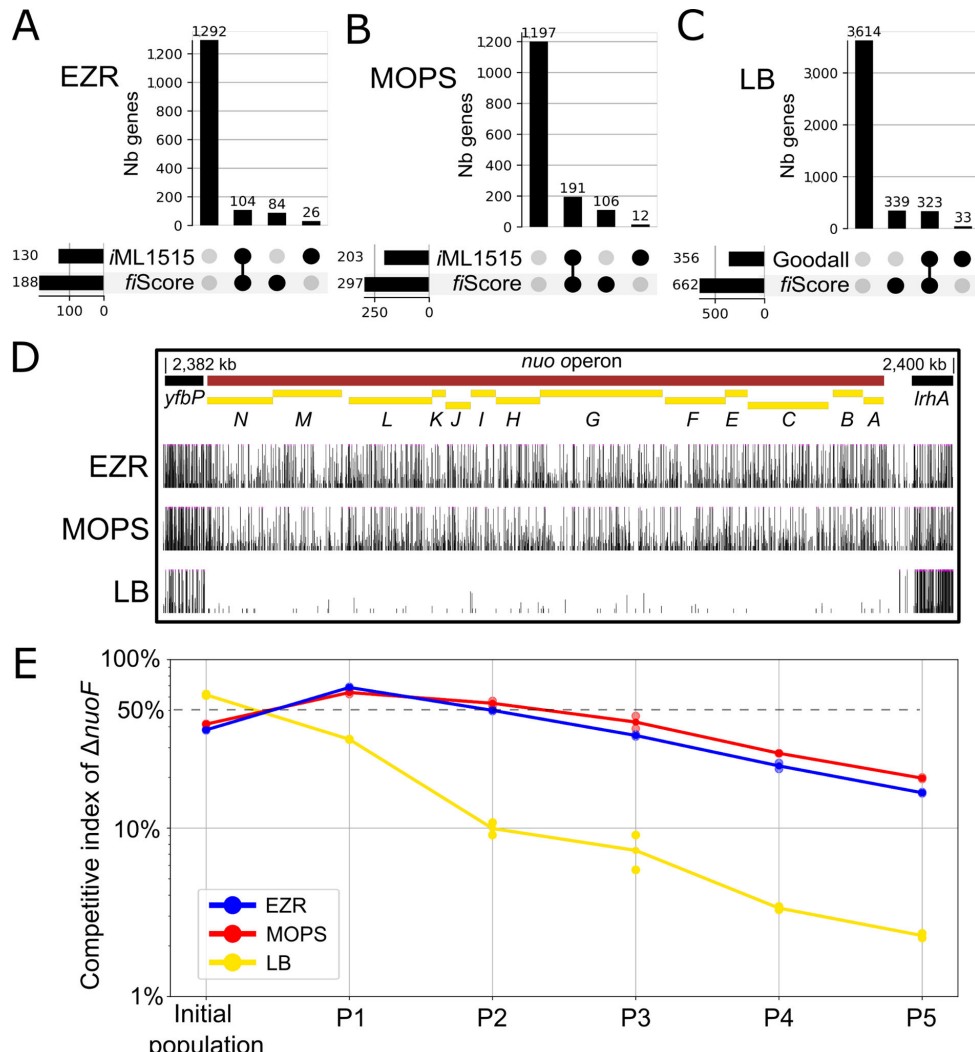

**FIG 3** Comparison of essentiality calls with existing data sets. (A and B) Comparison between genes predicted to have a significant impact on fitness by the *i*ML1515 model and our data in EZR (A) and MOPS (B). Black circles indicate Hi-FI or essential genes. (C) Comparison of LB Hi-FI gene calls determined in our study (fiScore) with previously published LB essential genes (E-Genes from Goodall et al. [14]). (D) Representative locus showing an operon (*nuo*) with differential fitness impact between tested culture media. (E) Competitive assay between *E. coli* BW25113 and BW25113Δ*nuoF* performed in all three media over five passages. The word "number" is abbreviated as "Nb."

understanding and highlights the need for further investigation into context-dependent gene essentiality.

## Functional landscape of core genes

Most of the Hi-FI genes are common across all three media (Fig. 4A and B). As expected, these core important genes are generally responsible for the basal functions of the cell required regardless of medium composition, such as genetic information processing or glycan production, based on the KEGG database classification (33) (Fig. 4C and D). Many metabolic genes are also found in the core set, for example, genes associated with the biosynthesis of lipids, which are not provided in any of the three tested media (Table S4). Interrogation of the EcoCyc database, a comprehensive and regularly updated source for gene characterization (5), revealed a more accurate count of 14 uncharacterized genes and 32 partially characterized genes, highlighting targets that warrant further investigation (Table S2). All 14 uncharacterized genes encode small polypeptides of up to

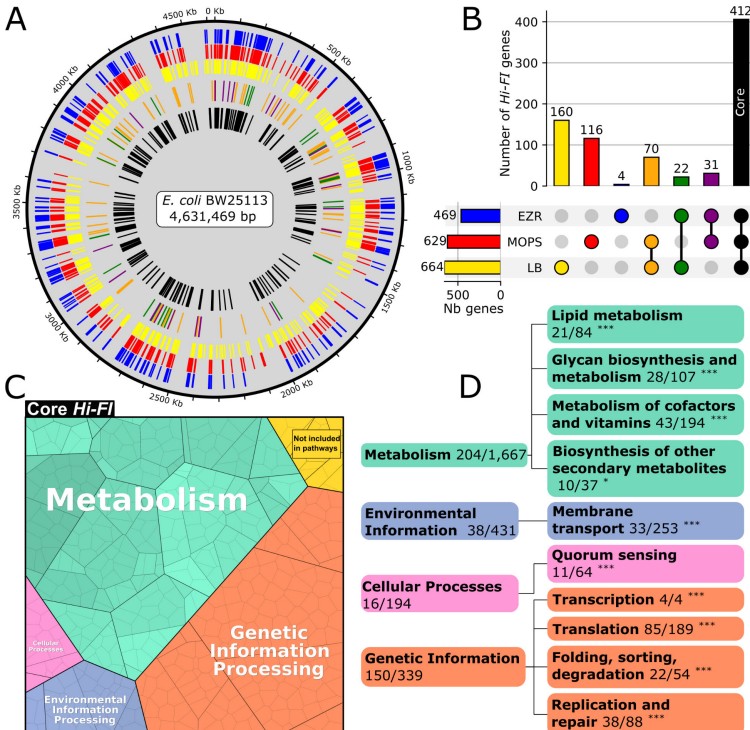

**FIG 4** Description of the core genome. (A) Representation of the Hi-FI genes on the *E. coli* BW25113 chromosome. Each color in the outer rings indicates fitness impact in a condition (EZR = blue, MOPS = red, LB = yellow), bars in the middle ring indicate fitness impact in two media (orange = MOPS and LB, green = EZR and LB, purple = MOPS and EZR), and black bars in the inner ring depict genes having a high fitness impact in all three conditions (core). (B) Upset chart representing the size of the different subsets of Hi-FI genes. (C) Repartition of the functions of the core genes into the main functional categories of the KEGG database represented using the Proteomaps tool (34) (D) Decomposition of the main functional categories of core genes. Each category is labeled with the number of genes belonging to this category found in the core and the total number of genes belonging to this category. The significance of ontology term enrichment compared to random expectation was tested using a hypergeometric distribution; *P*-values are indicated as follows: *, <0.05; **, <0.01; ***, <0.001. The word "number" is abbreviated as "Nb."

200 bp, while the partially characterized genes span a wide range of cellular functions. Examples include *tolQ*, associated with cell division, as well as *rimM* and *rimP*, which are involved in ribosome maturation. Having defined this core set of genes, we shifted our focus to a more granular analysis of fitness, examining genes whose medium fiScores suggest fitness contribution of specific gene sectors, as well as genes with moderate fitness impacts that allow for mutant persistence throughout the selection process.

## Detection of fitness-impacting gene sectors

To refine the detection of Hi-FI genes, we sought to identify genes that possess a sector important for fitness but that can tolerate insertions in the rest of their sequence (35). Such genes could have a relatively low fiScore but would still result in major fitness loss upon removal. We identified these genes by detecting stretches of contiguous MISS bins within genes exhibiting intermediate fiScores (above 15 and below 90), in combination with a clear split point where most bins are labeled HIT on one side and MISS on the other. Cases such as *ftsN*, *mqsA,* or *rne* (Fig. 5) are straightforward to interpret, with the important protein section matching known *ftsA* interaction domains (36), a toxin (MqsR) interaction domain (37), or the known RNA interaction domain. This shows that large sections in the 3′ of these genes are likely dispensable for the genes' functions in

these growth conditions. Other genes, such as *gpt*, which encodes a xanthine-guanine phosphoribosyltransferase, appear to have an important section only in EZR (Fig. 5C). We speculate that the role of its C-terminal region as a purine salvage enzyme is vital for competitiveness only in EZR.

Using an automated approach followed by manual curation, we identified 55 and 57 genes that harbor at least an important region that should be conserved for growth in EZR and MOPS media, respectively. Since the removal of these genes is expected to have a negative impact on fitness, they are considered Hi-FI genes when assembling the final list of genes required for robust growth in each medium. The complete list is available in Table S5.

## Identification of low fitness impact genes using temporal variation

Because of the competition inherent in HTTM experiments, an insertion mutant that carries even a slight fitness burden is expected to be gradually outcompeted by the rest of the population, which is reflected by a continual decline in associated read count (29). Conversely, an unaffected insertion mutant would have a constant or even increasing read count over the passages. Hi-FI genes identified using the fiScore approach are genes that have reached read count approaching background levels at P5 or earlier, reflecting a major impact on cell fitness upon inactivation (Fig. 1A). This approach, however, does not allow the detection of genes with lower fitness impact, whose read counts decline slowly over time but have not reached a background level by P5.

Using hierarchical clustering on the temporal variation of read abundance (Materials and Methods) from the non-Hi-FI genes in each medium, we produced five groups from which one displayed a clear depletion rate that would, therefore, be likely devoid of

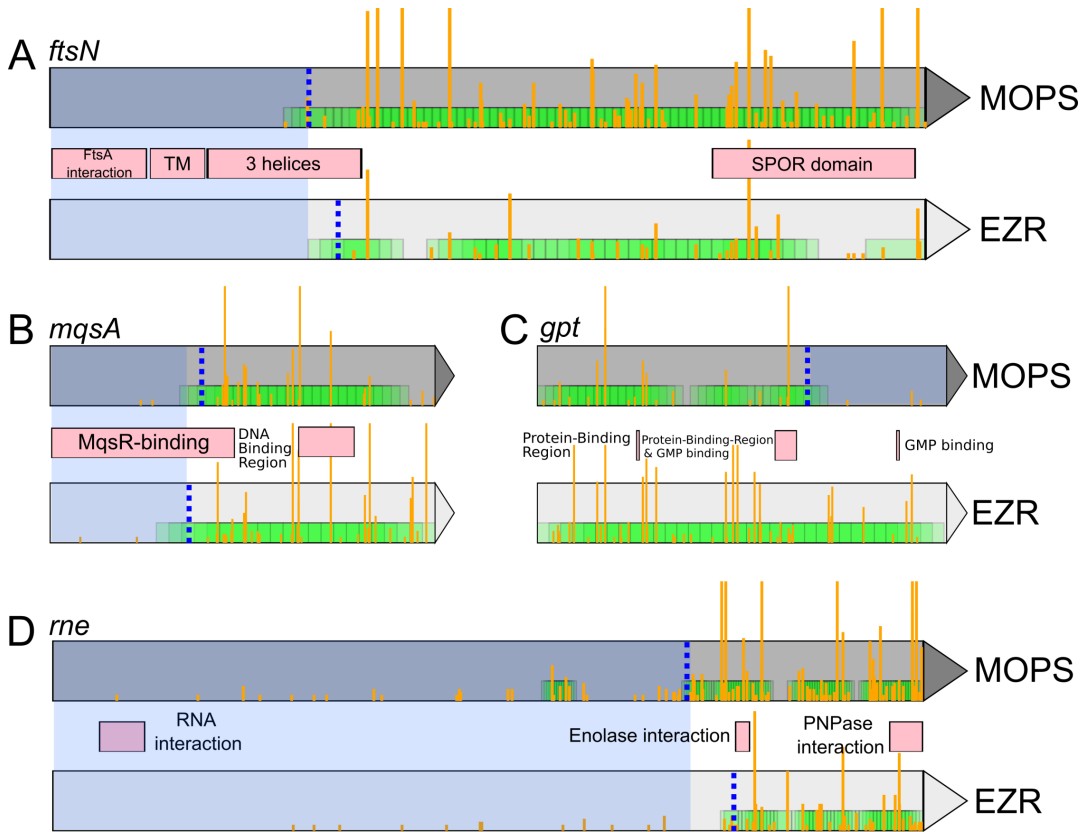

**FIG 5** Examples of genes with a fitness-impacting sector in MOPS or EZR media. (A–D) *ftsN, mqsA, gpt,* and *rne* genes, respectively. Yellow bars indicate transposon insertion sites, with height proportional to the number of mapped reads. The dashed blue line represents the optimal segregation between the HIT bin (green) and the important region (highlighted in blue). Pink boxes represent known domains and features (Uniprot database).

insertions after additional passages (Fig. 6; Fig. S6); we categorized this latter group as low fitness impact (Lo-FI) genes. Although the influence of each individual Lo-FI gene deletion is low, their cumulative removal may cause significant fitness hindrance. The rest of the genes were grouped under the broad no fitness impact (No-FI) category reflecting that their individual inactivation causes no significant negative effect on cellular fitness (Table S6). Temporal variation rates fluctuate widely among No-FI genes, ranging from nearly null (No-FI 1, 3, and 4) to strongly positive (No-FI 2). Those latter atypical cases might reflect an effective fitness gain upon inactivation or simply reflect an effect of the mere compositional nature of the data. Competition assays using single-gene mutants and a wild-type strain confirmed the vast majority of the expected fitness impact in the expected medium (Fig. S7).

Lo-FI genes were appended to the previously determined list of Hi-FI genes to form medium-specific gene modules containing the complete non-core gene set required for robust survival in either EZR or MOPS (Fig. 7A). These medium-specific gene "modules" are described in the following sections.

## EZR module

The EZR module is composed of 421 non-core genes, among which 93 are categorized as Hi-FI and 328 as Lo-FI in this medium (Fig. 7A; Table S7). As can be expected for a medium-specific gene module, the most significantly enriched KEGG terms in this module are all related to metabolism. The next most enriched terms being related to genetic information processing with a surprising prevalence of genes involved in DNA repair. Only four genes (*cycA, relA, argW,* and *rnlB*) are categorized as Hi-FI exclusively in this medium. Surprisingly, comparison with a list of known transporters in *E. coli* (38) revealed that no transporter other than *cycA* is categorized as Hi-FI in EZR, while 13 have a low fitness impact. One possible explanation for this observation is the functional redundancy of the transporters. For example, both *uraA* and *rutG* (39) genes are predicted to be responsible for uracil uptake but are considered No-FI in our screening. Meanwhile, the *upp* gene, which is required for uracil import by either one of those genes, is considered Hi-FI in both EZR and LB, the two media where uracil is available for import.

## MOPS module

In total, 656 genes were classified as Hi-FI for growth in MOPS medium, including the 412 core genes and 244 specifically important in MOPS. In addition, 779 genes were classified as Lo-FI in this medium. Together, these 2 groups define the MOPS module, comprising 1,013 non-core important genes (Fig. 7A; Table S7). Functional analysis of genes important for growth in MOPS revealed significant enrichment in metabolism-related genes, as expected from the composition of this minimal medium (Fig. 7B). Notably, pathways responsible for the biosynthesis of all 20 standard amino acids and all four nucleotides exhibited significant enrichment (Bonferroni-corrected *P*-value <0.05, Table S4). The mapping of the MOPS module gene set onto the KEGG "Biosynthesis of Amino Acids" metabolic pathway confirms that all reactions required to synthesize the 20 amino acids from central metabolic precursors are covered (Fig. S8). Notably, some of the reactions in these pathways can be performed by two or more genes, which should be individually less susceptible to inactivation. In each of these cases, one of the redundant genes is identified as a Lo-FI gene with the single exception of the last step of asparagine synthesis. This final step can be catalyzed by either AnsA or AnsB, two enzymes that appear capable of fulfilling the cell's asparagine requirement independently (40), without measurable loss of fitness.

As mentioned before, single-gene inactivation performed using the *i*ML1515 metabolic model under the MOPS condition (Fig. 3B) revealed a high degree of concordance between the two approaches (~92.1%). The 12 genes that were predicted as important for the fitness only by the model (*luxS, pabA, ubiD, ispA, bioH, panB, yrbG, zupT, hemG, hemL, pabB,* and *panC*) are all long and densely inserted genes. Those

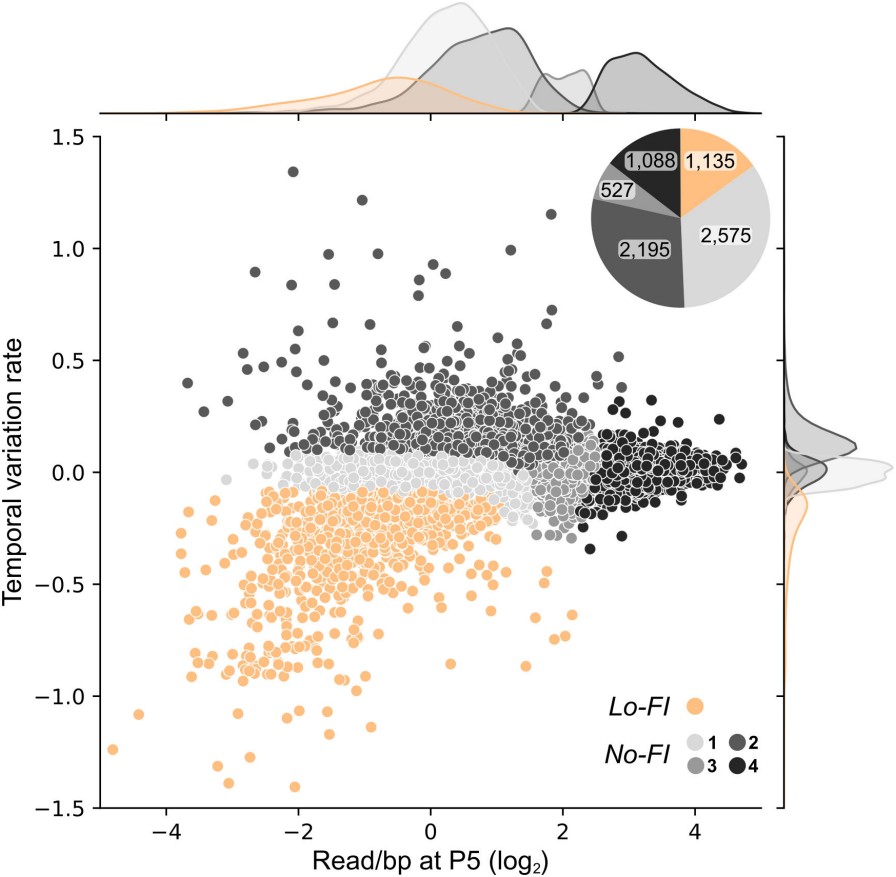

**FIG 6** Temporal variation in insertion counts uncovers fitness-affecting genes. Temporal variation rate of non-Hi-Fi genes in either defined medium is shown in function of their read count per base pair at P5. The rate is derived from the normalized read counts across passages. Values for EZR and MOPS are plotted as individual dots (for a total of 7,520 genes). Genes are colored according to hierarchical cluster assignment (Lo-Fi in orange, different types of No-Fi in different shades of gray) (Fig. S6). Marginal density plots along the axes show the distribution of each metric across gene categories. The inset pie chart displays the absolute gene count per category, both media combined.

genes might be functionally complemented by other mutants of the population or simply wrongly assigned as important. Among the hits not supported by the model, we identified some generic information-processing genes, such as *lysS* (tRNA synthetase). The high fiScore of this gene in defined media (100 in MOPS, 84.8 in EZR) suggests that its removal would severely impair the strain in these conditions, while having only a moderate impact in LB medium, where the fiScore is 25.6. Although *lysU* (fiScore = 0 in all three conditions) encodes an alternative lysyl-tRNA synthetase capable of supporting growth on its own (41), its expression in wild-type strains is generally extremely low and, therefore, insufficient to compensate for the inactivation of *lysS*. However, *lysU* is subjected to multiple regulatory mechanisms such as induction at elevated temperature (42), under anaerobic conditions (43), or low pH (44), and it is possible that one of these mechanisms, or an as-yet unidentified regulatory response, is specifically triggered in LB medium, thereby mitigating the fitness impact of *lysS* inactivation. As the import of charged tRNAs potentially present in LB is unlikely, given that to our knowledge no instance of tRNA or aminoacyl-tRNA import has been reported in prokaryotes, this effect is more plausibly attributed to differences in gene regulation rather than in the metabolizable components of the medium. This represents an interesting case in which medium-specific gene importance may be dependent on regulatory effects rather than on the ability to metabolize components of the medium.

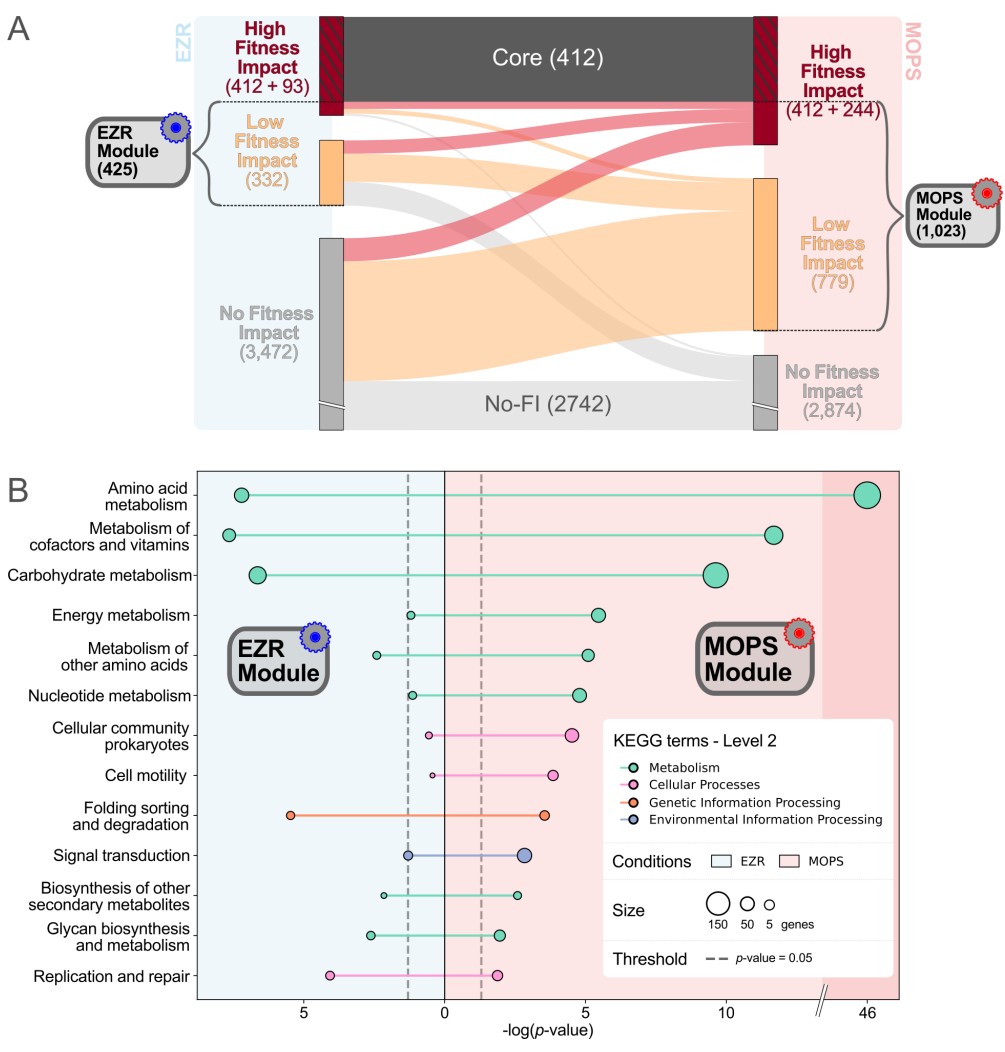

**FIG 7** Description of the medium-specific gene modules. (A) Sankey diagram showing gene status in EZR (left) and MOPS (right) media. Flows indicate changes in category of fitness impact between conditions, with colors (red, orange, and gray) reflecting the category in MOPS. Numbers denote the gene counts for each category and module. (B) Lollipop chart representing Level 2 KEGG ontology enrichment analysis for genes in the EZR (left) and MOPS (right) modules. The $-\log_{10}(P\text{-}$value) is plotted for each ontology term, with higher values indicating stronger enrichment and circle size as a function of the number of genes. Only ontology terms with a *P*-value below 0.05 are shown.

## DISCUSSION

The application of high-density transposon mutagenesis across 3 different growth conditions revealed a group of 412 common core genes, complemented by groups of medium-specific genes. The size and composition of this core group of fitness-impacting genes is consistent with other essentiality studies in *E. coli* (12, 14) and encode the main biological functions expected to be critical for bacterial life. On the other hand, we anticipated that the number of medium-specific genes would negatively correlate with the complexity of the medium. However, a total of 252 non-core Hi-FI genes were identified in LB, of which 160 are unique to this medium, significantly more than the 116 and 4 genes exclusive to MOPS and EZR, respectively (Fig. 4B). This is striking as LB is based on yeast extract, i.e., whole *Saccharomyces cerevisiae* lysate, and is assumed to be the most complete of the three tested media (45). However, optimal metabolization of LB is very complex, with cells sequentially depleting the most efficient amino acid available to harvest carbon (46). We hypothesize that this process would require many more genes

for a cell to be competitive in a growing population compared to one with a fixed and abundant carbon source, such as glucose. This supports the fact that the metabolism required for optimal proliferation in LB medium is highly intricate and fluctuates during growth, suggesting it may be ill-suited as a go-to medium for in-depth characterization, a point raised in several previous studies (26, 46, 47).

Overall, our results are consistent with those of other gene essentiality data sets, including other TIS-based studies (14) and metabolic models (32), while also providing additional refinement by identifying both high-fitness-impact gene sectors, and low-fitness-impact genes through temporal variation analysis.

When comparing our data set with the TIS study by Goodall et al. (14), most discrepancies involved smaller genes (Fig. 3C), which typically receive fewer insertions and are, therefore, more likely to fall just below or above significance thresholds by chance. Indeed, genes that differed between the two TIS data sets had a median size of 489 bp, compared to 810 bp for all annotated protein-coding genes (Wilcoxon rank-sum test: $Z = 8.76$, $P = 7.52e{-}22$). Some additional discrepancies arise when comparing our results to single-gene knockout data sets (12). For instance, certain genes classified as No-FI in our analysis (e.g., *ubiJ*, *alsK*, *bcsB*, *tnaB*) have also been reported as non-essential in other transposon-based studies (14), suggesting that the disagreement with single-gene knockouts may arise from methodological differences. Another difference to keep in mind is that competition-based assays can mask the fitness impact in some mutants that can compensate for gene inactivation through the uptake of trace nutrients (e.g., vitamins) released into the medium by other strains. Although strong competitive selection generally eliminates mutants with restricted access to essential metabolites (48), such cross-feeding may allow some otherwise non-viable mutants to persist.

Genes identified as Hi-FI in specific media are confirmed to have a major impact on fitness using competition assays with single-gene mutants in those same media (Fig. 7). For example, a Δ*tolQ* mutant, a core gene, displays a substantial loss of fitness in both tested media. In contrast, deletion mutants of genes considered to be medium-specific, such as *pheA* and *upp,* exhibit loss of fitness only in the expected medium. Unexpectedly, some genes classified as No-FI in both tested media, such as *pdeB*, also exhibit a very slight fitness reduction, approximately one less division over ~15 generations, compared to the parental strain. Because *pdeB* is relatively large (1,551 bp, among the top 15% largest genes), this discrepancy is unlikely to be due to gene size causing a false negative. This suggests that very minor fitness defects may remain undetected by our classification thresholds.

Regarding the fitness-impacting sectors (Fig. 5), an initial hypothesis was that occasional localized gaps of insertions could be caused by nucleoid-associated proteins (NAP), as suggested by a recent study (49). This phenomenon could have been responsible for MISS regions. However, the previously reported insertion-depleted areas typically associated with NAPs were not detected in our data set despite the presence of the associated genes (*hns*, *mukB*, *tus*, etc.). Insertion sites showing an abnormally high read count, another type of potential artifact, tend to be common across various samples in all three media, suggesting an advantageous insertion site during the competitive enrichment and/or a Tn5 transposase insertion bias, as previously reported (50). While the variety of sites that can receive transposons using Tn5 transposase is generally not limiting in TIS experiments, with nearly an insertion site every five base pairs in this study, exploring the precise *in vivo* insertion bias would be interesting. This includes factors such as NAPs, sequence specificity, and other more subtle causes, which could help improve the method by using this profile as a baseline to compare the post-selection insertion profiles.

Investigation of metabolic maps, such as the one for amino acid metabolism (Fig. S8) or the *uraA/rutG* redundancy, underscores that certain reactions can be catalyzed by more than one gene. This redundancy makes the fitness contribution of these genes hard to detect using approaches based on single-gene inactivation, such as single knockout or HTTM. To develop a comprehensive knowledge base for thorough and

predictive genomic simplification, a different approach involving whole-cell modeling (32, 51) or combinatorial gene inactivation (52) will need to be devised to predict potential cases of synthetic lethality.

In addition to the 16 uncharacterized genes among the core genes, there are an additional 146 genes with an uncharacterized status in the MOPS module and 60 in the EZR module (Table S7). Their varying fitness contribution across growth conditions could indicate a role in metabolism or its regulation. In addition, even well-characterized enzyme-encoding genes, although having an ostensibly clear function, may engage in moonlighting roles that are distinct from their annotated functions. This highlights the need for ongoing efforts to enhance gene characterization, even of a well-known organism like *E. coli*. Our medium-specific modules offer a practical design for precise, condition-specific engineering of this organism. Further investigations across a wider range of conditions, including different media and growth environments, would offer a more complete understanding of the specific contexts where each gene is important and help us learn more about their functions. This increased understanding will greatly enhance our ability to engineer microorganisms with specific capabilities for various biotechnological applications, while also expanding our fundamental knowledge about the function of numerous genes.

## MATERIALS AND METHODS

### Growth conditions

Cells were grown in LB broth (Miller), MOPS minimal medium + 0.2% glucose, or EZ-Rich + 0.2% glucose (EZR). LB was prepared from commercial powder (BioBasic SD7003) at the following concentrations: tryptone 10 g/L, NaCl 10 g/L, yeast extract 5 g/L. MOPS and EZR were prepared following published recipes (https://www.genome.wisc.edu/resources/protocols/mopsminimal.htm and https://www.genome.wisc.edu/resources/protocols/ezmedium.htm). Exact composition of all media is available in Table S3. When required during the selection process, spectinomycin was added to a final concentration of 100 µg/mL.

### Transposon mutagenesis and library preparation

Transposon mutagenesis experiments and the preparation of sequencing libraries were performed as previously published (15), with slight optimizations designed to increase efficiency. In short, a mini Tn5 transposon carrying a spectinomycin resistance gene was delivered to recipient *E. coli* cells via RP4-mediated conjugation using a mobilizable suicide plasmid. The donor strain tightly represses transposase expression to prevent premature transposition, ensuring insertions occur only in the target strain. After conjugation, mutants were cultured in one of three media (LB, MOPS, or EZR) and passaged every 24 h to allow selection against insertions in essential genes. At each time point, after the next passage was inoculated, remaining cells were pelleted, and genomic DNA was extracted. Library preparation was performed using a modified NEBNext Ultra II DNA Library Prep Kit, which enabled sequencing of genomic regions adjacent to transposon insertion sites in a similar fashion to Illumina Nextera libraries. The main optimizations were (i) increasing the number of donors during the conjugation by ~25%; (ii) changing the DNA polymerase used for the library amplification step for the SsoAdvanced Universal SYBR Green Supermix 2x (Bio-Rad); and (iii) adding an additional depletion step during the library preparation of early passages (P0 to P2) to remove contaminating mutagenesis plasmid sequences from the libraries. To do so, samples were treated with the AvrII restriction enzyme for 30 min at 37°C after the ligation of the adapters to selectively digest the plasmid. Samples were then processed as previously described. The complete updated protocols are available on Protocol.io: https://www.protocols.io/view/httm-transposon-mutagenesis-dd3428qw, https://www.protocols.io/view/httm-gdna-extraction-dd3828rw, and https://www.protocols.io/

view/httm-illumina-library-preparation-dd4d28s6. Sequencing was performed on either an Illumina or an Element AVITI platform, depending on instrument availability and pooling opportunities, using standard Illumina Nextera sequencing primers and generating paired-end reads of 50, 75, or 150 bp.

As a control, seven insertional mutants were isolated from an HTTM pool and were individually barcoded. Five of these mutants have only one major position that represents close to 95% of the reads, while dozens of artifactual positions are backed by a signal under 5% (Fig. S9). However, two mutants had their reads split evenly between two insertion positions, meaning that ~30% of mutants may have sustained two insertions in their genome simultaneously.

DNA extraction from LB-grown cultures consistently produced viscous lysates that clogged silica columns, reducing total DNA yield by ~80% in all three mutagenesis replicates. We expect that this technical limitation is the main bottleneck causing lower insertion counts in LB samples despite repeated attempts and alternative extraction methods.

## Read processing

Paired-end sequencing reads (PE 75) were first assessed for quality using FastQC (53) (v0.11.9) to evaluate per-base quality scores. Adapter sequences (ACTGTCTCTTATACACAT CT) were then trimmed from both read pairs using Cutadapt (54) (v4.9) and reads shorter than 10 nucleotides were discarded. Further quality filtering was performed using fastp (55) (v0.23.2), applying right-end trimming with a sliding window of size 4 and a mean quality threshold of 10. Read alignment was performed onto the reference genome using bwa mem from BWA (56) (v0.7.17) with a minimum seed length of 10. Output alignments were sorted and processed using samtools (57) (v1.4), which includes the removal of low-quality alignments (mapQ ≤10).

Insertion site identification was performed by processing alignment files to produce per-position insertion counts, with single-read insertions discarded as noise. Feature assignment was achieved by intersecting insertion data with annotated genomic features using Bedtools (58) (v2.30) while excluding features with low mappability based on Genmap (59) (v1.3). Gene-level insertion counts were computed by mapping insertion sites to genomic features. These profiles were then analyzed using either Bio-TraDIS Toolkit (21), which applies a statistical model based on prior distributions to classify genes as essential, ambiguous, or non-essential, or the sliding window analysis procedure described hereafter.

## Identification of high fitness impacted genes

To evaluate gene fitness impact, each gene was scanned with overlapping sliding windows (bins) whose length was determined by the average distance between genomic insertions. Since most (>80%) of the genes are expected to be non-essential, we evaluated the number of bins containing significantly fewer reads than the global average. Assuming that insertion sites are random and that the distances between insertions follow a Poisson distribution, we found that in a bin seven times wider than the average insertion distance, there is approximately a 3% chance of having two or fewer insertions by chance (Fig. S10). Considering that not all insertions have the same number of reads associated with them, we used the number of reads associated with each insertion as a proxy for the quality of the insertion to discriminate between true insertions and background noise. For each bin, the total number of aligned reads was compared to the expected number of reads, corresponding to twice the median number of reads per insertion site in the sample. When above this threshold, the bin was considered HIT and, thus, probably not important. This means that a bin with a single insertion with more than twice the median reads count could make a bin HIT on its own, but it would only affect the gene's score locally, while a bin with a few insertions having a single read each would be considered MISS. This approach provides resilience to both background noise and aberrantly high numbers of reads in single insertions that

can randomly happen during this kind of experiment. A graphical representation of the process is available in Fig. 2; Fig. S2 to S4. In rare cases where the gene size is smaller than the established bin size, the script is hardcoded to attempt at least five steps to roughly distinguish between full insertion coverage and occasional insertions.

## Detection of fitness-impacting gene sectors

Using the previously calculated bins, genes with a non-extreme fiScores (between 15 and 90) were scanned for stretches of quasi-consecutive HIT bins, meaning the longest stretch of HIT bin including at most a MISS bin. Genes exhibiting a streak of at least 20% of their bins as HIT were selected for manual curation and inclusion as either a fully or partially important gene. The limit between the fitness impacting region and the rest of the gene was determined as the bin having the highest ratio of HIT bins on one side and MISS bins on the other side.

## Temporal variation of insertion counts to identify fitness impacting genes

Prior to gene read counts per base pair (read/bp) calculation, each sample's read amount at every base position was capped at an upper threshold corresponding to the 95th percentile, therefore negating strong swinging effects of single highly covered positions. Passages 1 and 2 were excluded since they harbored sparser transposon densities and higher artifactual plasmid reads. A pseudocount equal to the minimal non-null value of the data set was added to the read/bp of each gene for each preserved passage (3, 4, and 5) in both defined media (EZR and MOPS) to avoid zeros. Values were then normalized using geometric means for size factor calculation. These normalized values were transformed into logarithmic fold changes ($log_2FC$) using the P3 timepoint as the origin, in a similar fashion as Gallagher et al. (29). The fold changes were used to approximate temporal variation rate ($log_2FC$/passage) via standard linear regression.

All non-Hi-FI genes (including genes containing a Hi-FI sector) were submitted to hierarchical clustering, using normalized read/bp that were clipped at the upper extreme percentile. Along with the temporal variation rate, they were standardized (mean-centered and scaled to unit variance). As a driving factor, the temporal variation rate was upweighted four times compared to the normalized read counts per base of the three passages. An embedding was produced using UMAP (60), specifically following the guidelines for clustering preprocessing (https://umap-learn.readthedocs.io/en/latest/clustering.html), which was then clustered using agglomerative clustering with SciPy's (61) default parameters. Following visual inspection of the agglomerative clustering dendrogram, a separation into five clusters was chosen. According to the temporal variation rate and final read count spectrum, a fitness label was attributed to each cluster (i.e., Hi-FI, Lo-FI, and No-FI). A visualization of the embedding with cluster labels is available in Fig. S6.

## Proteomaps

Proteomaps were generated using the Proteomap tool (34, 62) http://bionic-vis.biologie.uni-greifswald.de using a custom map template created from up-to-date (January 2025) E. coli KEGG Orthology data (https://www.kegg.jp/brite/ko00001). Unlike standard Proteomap templates where genes with multiple functions are assigned a function at random for plotting, genes with several functions were assigned suffixes (_1, _2, …) to allow mapping of all their known functions, and all known functions of a gene are displayed on the map using those aliases. The area assigned to each gene corresponds to the fiScore squared, allowing for a greater separation from genes having a low number of insertions.

## Competition assays

Competition assays were performed to measure the relative fitness of a specific single-gene deletion mutant from the Keio collection (12) against the BW25113 strain

containing a kanR-GFP cassette inserted in *lacZ* (15). For each tested medium, both the deletion mutant and the parental strain were precultured individually in the same medium, and an equivalent number of cells was mixed together based on their $OD_{600}$ values. Mixed populations were then used to inoculate duplicate wells of a 96 deep well plate containing 1.5 mL of the tested medium, with an initial $OD_{600}$ of ~0.005. Deep well plates were then incubated at 37°C and underwent a total of five daily passages under conditions similar to those of the HTTM experiment described at the beginning of the results section. To evaluate cell concentrations at the end of each passage, serial dilutions were performed in sterile 1× PBS, and the dilutions were fixed with a final concentration of 1% formaldehyde. Samples were then analyzed using a BD Accuri C6 Plus flow cytometer (BD Biosciences) equipped with a 488 nm laser. The FSC-H channel threshold was set at 2,500 to eliminate background noise, and the FL1-H (FITC) threshold was set at ~200 to discriminate between the two populations. Fluidics were set to high speed, and a maximum of 100,000 events or 40 µL were collected for each sample. Cell populations were segregated based on FITC signal, distinguishing the BW25113 kanR-GFP strain from its competitor.

## Gene characterization level and functional enrichment analysis

The gene characterization level was obtained from a specialized SmartTable from the EcoCyc website. Statistical overrepresentation of functional terms was calculated using the eco00001 Orthologs database from KEGG (https://www.kegg.jp/brite/ko00001) as a reference. The 09150 Organismal Systems and 09160 Human Diseases categories were omitted as they are not relevant in *E. coli* K-12 strains. The enrichment analysis was performed using the hypergeom function from the SciPy (61) Python library.

### *In silico* fitness prediction using metabolic model

Flux balance analysis was performed on the *i*ML1515 GEM using the COBRApy python module from the COBRA Toolbox (63) with simulated MOPS or EZR media. Genes were considered fitness impacting if their inactivation led to a predicted reduction in growth rate of more than 50%. Constraints used to simulate EZR and MOPS media are the same as described in a previous article troubleshooting the metabolism of a genome reduced strain in defined media (64) and are based on the estimated medium composition available in Table S3.

## ACKNOWLEDGMENTS

We are grateful to the Centre de Calcul Scientifique of the Université de Sherbrooke for technical assistance. Access to computational resources was provided in part by Calcul Québec (http://www.calculquebec.ca) and the Digital Research Alliance of Canada. We are grateful to Dominick Matteau, who assisted with data analysis and performed curation of the text and figures.

This work was funded by grants from the Fonds de recherche du Québec—Nature et technologies (FRQNT) #RN490760—486535 and the Natural Sciences and Engineering Research Council of Canada (NSERC) #2020-06328.

A.D.G. and A.C. performed all manipulations and sequencing. A.C. and S.J. wrote the manuscript together with the support of A.D.G., P.-É.J., and S.R. S.J. and A.C. performed the data analysis, assisted by M.M.S. A.C., S.R., A.D.G., S.J., and P.-É.J. designed the experiments and data analyses. P.-É.J., J.-P.C., and S.R. supervised and contributed their expertise and resources to the project. All authors revised the manuscript.

## AUTHOR AFFILIATIONS

[1]Département de Biologie, Université de Sherbrooke, Sherbrooke, Québec, Canada
[2]Centre de recherche du Centre hospitalier universitaire de Sherbrooke (CRCHUS), Sherbrooke, Québec, Canada

³Institut de recherche sur le cancer de l'Université de Sherbrooke (IRCUS), Sherbrooke, Québec, Canada

**AUTHOR ORCIDs**

Antoine Champie http://orcid.org/0000-0002-4628-5895
Simon Jeanneau http://orcid.org/0000-0003-1907-2832
Amélie De Grandmaison http://orcid.org/0009-0005-4286-5461
Mathias Martin Silva http://orcid.org/0009-0001-6363-5084
Pierre-Étienne Jacques http://orcid.org/0000-0002-3961-294X
Sébastien Rodrigue http://orcid.org/0000-0002-5366-7234

**FUNDING**

| Funder | Grant(s) | Author(s) |
|---|---|---|
| Fonds de recherche du Québec – Nature et technologies | #RN490760 - 486535 | Jean-Philippe Coté |
| | | Pierre-Étienne Jacques |
| | | Sébastien Rodrigue |
| Natural Sciences and Engineering Research Council of Canada | #2020-06328 | Jean-Philippe Coté |
| | | Pierre-Étienne Jacques |
| | | Sébastien Rodrigue |

**AUTHOR CONTRIBUTIONS**

Antoine Champie, Conceptualization, Data curation, Formal analysis, Investigation, Methodology, Validation, Visualization, Writing – original draft, Writing – review and editing | Simon Jeanneau, Conceptualization, Data curation, Formal analysis, Software, Visualization, Writing – original draft, Writing – review and editing | Amélie De Grandmaison, Data curation, Investigation, Methodology | Mathias Martin Silva, Visualization | Jean-Philippe Coté, Funding acquisition, Supervision, Writing – review and editing | Pierre-Étienne Jacques, Conceptualization, Data curation, Formal analysis, Investigation, Project administration, Software, Supervision, Validation, Writing – original draft, Writing – review and editing | Sébastien Rodrigue, Conceptualization, Data curation, Formal analysis, Resources, Supervision, Validation

**DATA AVAILABILITY**

Raw reads of the HTTM experiments available as fastq files in the NCBI Sequence Read Archive under accession number: PRJNA1300301. A trackhub displaying all insertion sites at every timepoint for each medium is available at the following link: https://genome.ucsc.edu/cgi-bin/hgHubConnect?hgHub_do_redirect=on&hgHubConnect.remakeTrackHub=on&hgHub_do_firstDb=1&hubUrl=https://g-f2b62d.6d81c.5898.data.globus.org/Champie_2025/Champie_2025.hub.txt).

**ADDITIONAL FILES**

The following material is available online.

Supplemental Material

**Supplemental material (mSystems01425-25-s0001.pdf).** Supplemental figures and captions for supplemental tables.
**Table S1 (mSystems01425-25-s0002.xlsx).** Number of reads and mapped insertions for each time point.
**Table S2 (mSystems01425-25-s0003.xlsx).** fiScore of all genes in all 3 media.
**Table S3 (mSystems01425-25-s0004.xlsx).** Fitness prediction made using the iML1515 metabolic model.

**Table S4 (mSystems01425-25-s0005.xlsx).** Enriched KEGG functions in each primary important group of genes and in the modules.

**Table S5 (mSystems01425-25-s0006.xlsx).** Description of genes harboring a fitness-impacting sector.

**Table S6 (mSystems01425-25-s0007.xlsx).** Gene summary.

**Table S7 (mSystems01425-25-s0008.xlsx).** All genes composing both modules.

## Open Peer Review

**PEER REVIEW HISTORY (review-history.pdf).** An accounting of the reviewer comments and feedback.

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
