## [Reviewer comments · mSystems]

Comparative Analysis of Gene Importance in *Escherichia coli* Across Growth Conditions

Antoine Champie, Simon Jeanneau, Amelie De Grandmaison, Mathias Martin Silva, Jean-Philippe Coté, Pierre-Étienne Jacques, and Sébastien Rodrigue

Corresponding Author(s): Sébastien Rodrigue, Université de Sherbrooke

Review Timeline:

Submission Date:	October 5, 2025
Editorial Decision:	October 30, 2025
Revision Received:	January 23, 2026
Editorial Decision:	February 13, 2026
Revision Received:	March 2, 2026
Accepted:	March 5, 2026

Editor: Julia Willett

Reviewer(s): The reviewers have opted to remain anonymous.

Transaction Report:

DOI: <https://doi.org/10.1128/msystems.01425-25>

Re: mSystems01425-25 (Comparative Analysis of Gene Importance in Escherichia coli Across Growth Conditions)

Dear Prof. Sébastien Rodrigue:

As you will see, the reviewers highlighted several strengths of the manuscript and potential impact on the field, as well as comments that should be addressed in a revised manuscript. In addition to carefully considering the reviewer comments, please also address any instances of non-specific language (such as precarious or important) and use more precise terms (such as fitness or quasi-essential) where possible. Please also ensure that, when possible, any shared data is available through robust, stable databases to ensure long-term access to interactive datasets.

Revision Guidelines

Sincerely,
Julia Willett
Editor
mSystems

Reviewer #1 (Comments for the Author):

The manuscript by Champie, Jeanneau & De Grandmaison et al. describes the construction of transposon mutant libraries in *E. coli* K-12 strain BW25113 and the growth of these libraries in 3 different media to analyse the contribution of each gene to fitness. Altogether they identify a core set of genes that are required for growth in all conditions, in addition to media-specific important genes. They also segregate genes qualitatively based on depletion rate into three further groups: secondary important genes, precarious (ambiguous) genes, and neutral genes that can be removed with minimal impact, to define gene modules that would be beneficial for growth in each condition. This distinction enables the differentiation of primary and secondary importance of genes with biosynthetic functional overlap.

They present a new window-scanning analysis method, which conceptually has been applied before but such tools are comparatively lacking for analysis of Tn5 data compared to mariner transposon datasets. Publication of their analysis method would be a welcome tool for the field.

Overall the manuscript is clear and well written. The authors are rigorous in benchmarking their sliding-window analysis method against published datasets and provide extensive analysis of their data. Their data provide new insights into the metabolic requirements of *E. coli* for growth in defined media and their work contributes towards a wider understanding of the synthetic genetic requirement for a minimal genome derived from *E. coli*.

My comments are mostly minor, with 2 points surrounding gene length and growth rate/insertion density for further comment from the authors.

Minor comments

line 54 - transposon mutants

line 87-88 - There are a couple of studies that demonstrate this point, e.g. Rousset et al. (PMID: 33462433) and Poulsen et al. (PMID: 31036669)

line 112 - referred to

line 132 - the insertion numbers here match those for passage 5 in S1 table for EZR, but not MOPS or LB (601505 and 229591 respectively)

line 147 Fig. 1 - purple (ackA and pta) is important in red and blue [EZR and MOPS]? red (cvpA) is red [MOPS] only? needs adding to the legend

line 155-6 - Do the authors mean reads or insertions per gene rather than per base?

line 169 - because gene importance is determined by the proportion of bins (of a fixed size) classed HIT or MISS in a given gene, can the authors comment on the effect this has for classification of small genes? how are short genes impacted? between replicate data, are smaller genes more prone to variation in iScore classification?

Given the localised 9 bp duplication during tn5 insertion, enabling transposon insertion at the extreme 5' and 3' end of essential genes, (and in some case as much as 20% of the 3' end is dispensable in essential genes PMID: 39207104), are smaller essential genes more likely to be wrongly classified using this sliding-window approach?

Lastly, how are iScores impacted by ori/ter insertion bias (if any)?

line 219 - S. Table 2 characterization data - lolB listed as uncharacterized, and bamD, ftsL, ftsQ etc. as partial, but the function of these genes is well documented.

Figure 6 - what do the data look like when separated by mediums? And should this be provided as a supplementary figure? Given the greater number of primary and secondary genes contributing to fitness in MOPS, was the overall rate of depletion faster?

Can the authors comment on doubling time of the parent strain in each medium, and how many generations a (neutral) cell would go through during a single passage? Does growth rate correlate with selection strength and depletion rate?

line 328-336 - Do the authors have an explanation for consistently fewer insertions overall in the LB datasets? Is this the same for each replicate, that unique insertions never get above 200,000? (despite more dense libraries reported here and elsewhere) and if so, is this a caveat for making broth libraries in rich media?

Many of the LB-only important genes are known "slow growers" (in LB), so it's entirely reasonable that they would be outcompeted during competitive outgrowth, however, it's hard to interpret whether the higher number of LB-specific important genes are due to selection pressure, or whether this is a false positive inflation caused by a more sparse dataset to begin with. Given that the authors have a temporal dataset, it might be helpful to know how some of the LB-specific genes compare between passages.

line 344 - It is not clear which genes are being referred to here by 'entire' - is this the median gene size for the "entire important genes" and "entire essential genes"?

line 356-358 1551 bp is not a small gene?

Line 517 - Author D.M is not listed?

Document only gives references up to no. 37. Beyond this could not be reviewed.

Reviewer #2 (Comments for the Author):

Summary:

Champie et al. propose a new way to determine gene essentiality using a traditional transposon-seq approach in *E. coli*. They compare 3 different types of media; two are defined (MOPS and EZR) and one undefined (LB). Using this approach they define sets of genes that are essential in all three media conditions as well as genes that are specific to one or two of the media. Because they serially passage the transposon library through 5 rounds of growth (through mid-log and stationary phase), they are able to select against genes that are essential at any point during growth.

Based on the number of transposon insertions, they can classify genes as essential, somewhat essential, and non-essential. Instead of just looking at the total number of insertions per gene, they use a sliding window approach that is less sensitive to gene length. Employing the use of a third class of genes (i.e., the somewhat essential genes that impact overall fitness) helps to create a more nuanced view of the role of genes during growth.

Comments:

Line 35: Suggest "possible" instead of "accessible".

Line 42: Suggest "less of an" instead of "a lighter". As a general note throughout, there are several more colloquial, subjective adjectives (i.e., "lighter", "precarious") used. It would be best to remove these and/or make sure that they are quantifiable.

Line 81: "providing a unique perspective on the question" is awkwardly worded. Do you mean that each strategy has different advantages and pitfalls. If so, it may also help to list some of these and how/why your strategy ameliorates some of the weaknesses or builds on the strengths.

Line 118: I disagree with introducing the terminology of "important" over something more traditional like "fitness" genes, which should still capture the need for the genes at different growth phases. Although the screen done here looks at three in vitro media conditions, it could be argued that virulence genes may be quite important during infection, but non-essential in laboratory culture. As the *E. coli* strain used is a laboratory strain and most of the genes that had insertions tended to involve central carbon metabolism, "fitness" still makes more sense to me.

Line 128: Please include the enzyme name here and not just in the methods.

Line 130: Please define "acceptable level". What was the fold-change reduction?

Line 141: It was unclear when the cultures were actually passed each time. Was stationary phase defined as overnight? A specific number of hours? An OD reading? This is especially important to note given that three different media are used. I didn't see this information in the methods either.

Line 172: How many Tn insertions do you estimate occur after 5 passages? How would you account for synthetic lethality phenotypes? This is touched on briefly in the discussion, but would like to see an estimate of the number of insertions per strain after continuous growth for 5 passages.

Line 183 (Fig 2C): For this and subpanels like it throughout the manuscript, what is the x-axis or the 4 categories that are represented?

Line 187: Why do you think this is true? I would imagine that there is less variation in the production and lots of salts and sugars compared to how yeast extracts are manufactured. Are there alternative explanations?

Line 195: What was the number of previously known genes rather than "most"? Is there a succinct way to describe what number and types of genes are meant by "additional genes"?

Line 197: Please include a short descriptor of what the *nuo* operon and *nuoF* encode. Would you expect a mostly amino acid/peptide media like LB to require this more so than the more carbohydrate heavy media?

Line 216: Here and throughout (although this gets better in the second half of the manuscript), please give examples of genes- especially if it is a manageable number like 2.

Line 218: Define "accurate". How do you know what the ground truth is?

Line 225: Instead of "color matching" please just list out what the colors correspond to. It took a while to understand that you mixed the two colors assigned to the media. Great way to help remember it when looking at the figure, but hard to interpret as worded in the legend.

Line 232: Please define where (i.e., what media conditions) the numerator and denominator represent in the legend.

Figure 6: Here, and in supplemental, what is the difference between the three neutral/gray categories?

Lines 302-304: Where do most of the insertions occur? If most are in the beginning of the gene, you might expect to see a lethal phenotype, but potentially if they are mostly toward the end and the *E. coli* is growing in a rich media, the abundance of lysine may be able to compensate for slowed charging, whereas it may not be able to in minimal media and/or if it were relying on making its own lysine from scratch.

Line 325: Suggest changing "coherent" with "consistent".

Line 357-358: Please reword or delete the awkward phrasing at the end of the sentence to improve clarity.

Line 399: Although it is described elsewhere and you link to protocols.io, please include a brief summary of what transposon was used and how the mutagenesis was performed.

Line 415: Please include the GenBank/ENA accession number used.

Line 478: Some words are missing: "was lower than".

Line 480: Please include a brief description of what is in the reference.

Line 496: How were 5 clusters decided upon? Statistically? Some biological reason?

Sup. Fig 1: Since this data is based off of 20 individual cultures per medium, suggest showing number of mutations per culture as dots and adding a standard deviation.

Sup. Fig 2: Suggest changing the box that says "Table_S2" to something more descriptive, since this is a schematic.

Summary:

Champie et al. propose a new way to determine gene essentiality using a traditional transposon-seq approach in *E. coli*. They compare 3 different types of media; two are defined (MOPS and EZR) and one undefined (LB). Using this approach they define sets of genes that are essential in all three media conditions as well as genes that are specific to one or two of the media. Because they serially passage the transposon library through 5 rounds of growth (through mid-log and stationary phase), they are able to select against genes that are essential at any point during growth.

Based on the number of transposon insertions, they can classify genes as essential, somewhat essential, and non-essential. Instead of just looking at the total number of insertions per gene, they use a sliding window approach that is less sensitive to gene length. Employing the use of a third class of genes (i.e., the somewhat essential genes that impact overall fitness) helps to create a more nuanced view of the role of genes during growth.

Comments:

Line 35: Suggest “possible” instead of “accessible”.

Line 42: Suggest “less of an” instead of “a lighter”. As a general note throughout, there are several more colloquial, subjective adjectives (i.e., “lighter”, “precarious”) used. It would be best to remove these and/or make sure that they are quantifiable.

Line 81: “providing a unique perspective on the question” is awkwardly worded. Do you mean that each strategy has different advantages and pitfalls. If so, it may also help to list some of these and how/why your strategy ameliorates some of the weaknesses or builds on the strengths.

Line 118: I disagree with introducing the terminology of “important” over something more traditional like “fitness” genes, which should still capture the need for the genes at different growth phases. Although the screen done here looks at three in vitro media conditions, it could be argued that virulence genes may be quite important during infection, but non-essential in laboratory culture. As the *E. coli* strain used is a laboratory strain and most of the genes that had insertions tended to involve central carbon metabolism, “fitness” still makes more sense to me.

Line 128: Please include the enzyme name here and not just in the methods.

Line 130: Please define “acceptable level”. What was the fold-change reduction?

Line 141: It was unclear when the cultures were actually passed each time. Was stationary phase defined as overnight? A specific number of hours? An OD reading? This is especially important to note given that three different media are used. I didn't see this information in the methods either.

Line 172: How many Tn insertions do you estimate occur after 5 passages? How would you account for synthetic lethality phenotypes? This is touched on briefly in the discussion, but would like to see an estimate of the number of insertions per strain after continuous growth for 5 passages.

Line 183 (Fig 2C): For this and subpanels like it throughout the manuscript, what is the x-axis or the 4 categories that are represented?

Line 187: Why do you think this is true? I would imagine that there is less variation in the production and lots of salts and sugars compared to how yeast extracts are manufactured. Are there alternative explanations?

Line 195: What was the number of previously known genes rather than "most"? Is there a succinct way to describe what number and types of genes are meant by "additional genes"?

Line 197: Please include a short descriptor of what the *nuo* operon and *nuoF* encode. Would you expect a mostly amino acid/peptide media like LB to require this more so than the more carbohydrate heavy media?

Line 216: Here and throughout (although this gets better in the second half of the manuscript), please give examples of genes—especially if it is a manageable number like 2.

Line 218: Define "accurate". How do you know what the ground truth is?

Line 225: Instead of "color matching" please just list out what the colors correspond to. It took a while to understand that you mixed the two colors assigned to the media. Great way to help remember it when looking at the figure, but hard to interpret as worded in the legend.

Line 232: Please define where (i.e., what media conditions) the numerator and denominator represent in the legend.

Figure 6: Here, and in supplemental, what is the difference between the three neutral/gray categories?

Lines 302-304: Where do most of the insertions occur? If most are in the beginning of the gene, you might expect to see a lethal phenotype, but potentially if they are mostly toward the end and the *E. coli* is growing in a rich media, the abundance of lysine may

be able to compensate for slowed charging, whereas it may not be able to in minimal media and/or if it were relying on making its own lysine from scratch.

Line 325: Suggest changing “coherent” with “consistent”.

Line 357-358: Please reword or delete the awkward phrasing at the end of the sentence to improve clarity.

Line 399: Although it is described elsewhere and you link to protocols.io, please include a brief summary of what transposon was used and how the mutagenesis was performed.

Line 415: Please include the GenBank/ENA accession number used.

Line 478: Some words are missing: “was lower than”.

Line 480: Please include a brief description of what is in the reference.

Line 496: How were 5 clusters decided upon? Statistically? Some biological reason?

Sup. Fig 1: Since this data is based off of 20 individual cultures per medium, suggest showing number of mutations per culture as dots and adding a standard deviation.

Sup. Fig 2: Suggest changing the box that says “Table_S2” to something more descriptive, since this is a schematic.

Response to reviewers' comments

We sincerely thank the reviewers for their time, effort, and constructive comments. Their thoughtful feedback has greatly strengthened the clarity and rigor of our manuscript. Below, we provide detailed responses to each point raised. For ease of reading, the reviewers' comments are shown in blue, and our responses appear immediately below in black. In addition to the requested changes, we have clarified numerous sentences and made cosmetic changes to some of the figures in order to better uniformize colors and graphs across the articles. We believe these modifications have significantly improved the overall quality of the manuscript.

Reviewer #1

Summary:

The manuscript by Champie, Jeanneau & De Grandmaison et al. describes the construction of transposon mutant libraries in *E. coli* K-12 strain BW25113 and the growth of these libraries in 3 different media to analyse the contribution of each gene to fitness. Altogether they identify a core set of genes that are required for growth in all conditions, in addition to media-specific important genes. They also segregate genes qualitatively based on depletion rate into three further groups: secondary important genes, precarious (ambiguous) genes, and neutral genes that can be removed with minimal impact, to define gene modules that would be beneficial for growth in each condition. This distinction enables the differentiation of primary and secondary importance of genes with biosynthetic functional overlap.

They present a new window-scanning analysis method, which conceptually has been applied before but such tools are comparatively lacking for analysis of Tn5 data compared to mariner transposon datasets. Publication of their analysis method would be a welcome tool for the field.

Overall the manuscript is clear and well written. The authors are rigorous in benchmarking their sliding-window analysis method against published datasets and provide extensive analysis of their data. Their data provide new insights into the metabolic requirements of *E. coli* for growth in defined media and their work contributes towards a wider understanding of the synthetic genetic requirement for a minimal genome derived from *E. coli*.

My comments are mostly minor, with 2 points surrounding gene length and growth rate/insertion density for further comment from the authors.

We are grateful to Reviewer #1 for the care and attention devoted to evaluating our manuscript. The thoughtful suggestions prompted meaningful improvements throughout the manuscript. We hope that the modification and improvements adequately address the referee's comments and contribute to a clearer and stronger manuscript.

Comments:

Minor comments

line 54 - transposon mutants

- The text was correct as suggested.

line 87-88 - There are a couple of studies that demonstrate this point, e.g. Rousset et al. (PMID: 33462433) and Poulsen et al. (PMID: 31036669)

- We now cite these articles as suggested.

line 112 - referred to

- The text was correct as suggested.

line 132 - the insertion numbers here match those for passage 5 in S1 table for EZR, but not MOPS or LB (601505 and 229591 respectively)

- This oversight has been corrected in both the text and Figure 1. After verification, the valid filtered counts from Table S1 have replaced the previously unfiltered values in the text. This comment also prompted us to validate other reported values, such as insertion density, and we confirm that they are accurate. Thank you for bringing this to our attention.

line 147 Fig. 1 - purple (ackA and pta) is important in red and blue [EZR and MOPS]? red (cvpA) is red [MOPS] only? needs adding to the legend

- The legend has been revised to specify all the colors used in the figure.

line 155-6 - Do the authors mean reads or insertions per gene rather than per base?

- The intended meaning of the sentence was insertions per base pair within each gene. However, we have revised the wording to read/insertions per gene, as this metric is closely related, subject to the same limitations discussed, and is more familiar and widely used outside our research group.

line 169 - because gene importance is determined by the proportion of bins (of a fixed size) classed HIT or MISS in a given gene, can the authors comment on the effect this has for classification of small genes? how are short genes impacted? between replicate data, are smaller genes more prone to variation in iScore classification?

- Small genes are indeed more challenging to analyze with this approach, as common for any Tn-seq analysis method (now discussed in **Section** : Discussion, paragraph #2). When a gene's size approaches the bin size, the fitness score becomes less granular because fewer bins can fit within the gene. Towards the extreme low end, genes similar in size to a single bin are almost classified in a binary manner. In rare cases where the gene size is smaller than the bin size, the script is hardcoded to attempt at least five steps to roughly distinguish between full insertion coverage and occasional insertions. (**Section** : Identification of fitness impacted genes, last sentence)
- Reads were always processed after pooling replicates to maximize insertion site coverage. Since the method performs best with at least ~200,000 insertions for reliable results, most individual replicates would not provide sufficient coverage for this type of analysis. To showcase the robustness of the approach, we randomly divided reads from all replicates of the passage 5 in EZR medium into three mini-pools of similar insertion density (~300,000 insertions each) and calculated iScores (named *fiScores* in the revised version) using the same pipeline applied throughout the study. We then evaluated the variation in fiScore for each gene across these three mini-pools as shown in the Figure below. Genes are ordered by size on the X-axis, and fiScore values are shown on the Y-axis. Variation

categories are color-coded: Gold: <5, Green: 5-15, Blue: 15-30, Red: >30.

- Overall, 75.7% of genes exhibited <5 *fiScore* variation, 19.6% fell within 5-15, 3.4% within 15-30, and only 1.3% exceeded 30. These results indicate that the vast majority of genes (>95%) remain highly consistent across replicates even when using downsized datasets. Genes with higher variability are highly enriched under 100 bp. As those mini-pool insertion density is half of the full dataset, we expect this lower limit to be reduced even further.

Given the localised 9 bp duplication during *tn5* insertion, enabling transposon insertion at the extreme 5' and 3' end of essential genes, (and in some case as much as 20% of the 3' end is dispensable in essential genes PMID: 39207104), are smaller essential genes more likely to be wrongly classified using this sliding-window approach?

- We thank the reviewer for this interesting article, it is a great support for the "Detection of fitness-impacting gene sectors" section where it is now cited. As previously noted, small genes are the known weak point of any transposon-based method and are more likely to be misclassified. Mention of this phenomenon is now included in the discussion (paragraph #2). In an attempt to mitigate this insertion bias, insertions at gene extremities are counted in fewer bins compared to insertions in the core region. Since we are using overlapping sliding windows, insertions near the start and end of a gene are taken into account only 1-4 times, whereas insertions in the middle are considered 5 times (given the 20% step size used in this study). Moreover, to address cases where ends are dispensable, we complemented the *fiScore* analysis with the detection of important regions, as illustrated in Figure 5.

Lastly, how are *iScores* impacted by *ori/ter* insertion bias (if any)?

- We did not observe a strong impact of insertion bias on read distribution at passage 5, as can be visualized on the provided TrackHub screenshot.

- Furthermore, as can be seen on the graph below, showing the genome-wide plots of *fiScores* (Y-axis) across the genome (X-axis) in each medium (one color per medium) do not indicate any systematic bias.

line 219 - S. Table 2 characterization data - *lolB* listed as uncharacterized, and *bamD*, *ftsL*, *ftsQ* etc. as partial, but the function of these genes is well documented.

- The gene characterization level was obtained from a specialized SmartTable from the EcoCyc website. At the time of analysis, our local copy of this SmartTable indicated that some genes, such as *lolB* and *bamD*, were uncharacterized. Given that *lolB* is indeed well characterized (and has been for quite some time according to the gene entry), we are unsure why our relatively recent (~1 year old) version of the table contained inaccurate information but we updated the table and redid the analysis. Other genes, such as *ftsL* and *ftsQ*, are still listed as partially characterized in the current database. Since we do not have specific expertise regarding these genes and the definition of “well” versus “partially” characterized can vary, we chose to defer to the EcoCyc classification. We have revised the manuscript and supplementary tables to include the latest EcoCyc data available, in which *lolB* and a few other genes are now considered “characterized”.

Figure 6 - what do the data look like when separated by mediums? And should this be provided as a supplementary figure?

EZR

MOPS

- We did plot the figures for separated media as shown below, but considering that the differences are subtle, we do not plan to include them in the supplementary material. As in Supplementary Figure 6, we also included the category of Primary genes (now renamed *Hi-FI* following a suggestion from reviewer #2), but zoomed on the central region as in main Figure 6.

Given the greater number of primary and secondary genes contributing to fitness in MOPS, was the overall rate of depletion faster?

Can the authors comment on doubling time of the parent strain in each medium, and how many generations a (neutral) cell would go through during a single passage? Does growth rate correlate with selection strength and depletion rate?

- Based on the fact that the daily dilution factor was relatively low ($\sim 1/15$) across all tested media, allowing cells to undergo approximately four doublings before reaching the original optical density, we estimate that the rate of depletion is similar in MOPS vs the other media. This limited dilution is required to enable working with small volumes while maintaining sufficient mutant diversity, minimizing the risk of randomly transferring only a subset of the population during each passage. The parent strain's doubling times in deep-well plates are approximately 38 minutes in LB, 45 minutes in EZR, and 50 minutes in MOPS, and after 24 hours of growth, cultures in all media have long reached their maximum cell density per tube. Therefore, we estimate that the overall capacity for growth, and by extension the strength of selection and rate of depletion should be comparable across media.

line 328-336 - Do the authors have an explanation for consistently fewer insertions overall in the LB datasets? Is this the same for each replicate, that unique insertions never get above 200,000? (despite more dense libraries reported here and elsewhere) and if so, is this a caveat for making both libraries in rich media?

- The main issue encountered was technical: DNA extraction from cultures grown in LB medium consistently caused clogging of the silica columns used in our protocol, systematically resulting in an average reduction of total DNA yield by $\sim 80\%$ in each replicate we tested. Although, in theory, sufficient DNA should be present given that each mutant exists in multiple copies within the population, the LB samples consistently produced lower insertion counts. This issue is now mentioned

in the *Transposon mutagenesis and library preparation* section of the Material and methods.

- We repeated the mutagenesis three times and tested alternative extraction methods, but these efforts did not resolve the issue. Since our primary interest was in defined media, and the final iteration of the protocol yielded sufficient insertions at passage 5 (P5) for bin-based analysis, we chose to focus our efforts on time-resolved analysis in the defined media.

Many of the LB-only important genes are known "slow growers" (in LB), so it's entirely reasonable that they would be outcompeted during competitive outgrowth, however, it's hard to interpret whether the higher number of LB-specific important genes are due to selection pressure, or whether this is a false positive inflation caused by a more sparse dataset to begin with. Given that the authors have a temporal dataset, it might be helpful to know how some of the LB-specific genes compare between passages.

- We thank the reviewer for this insightful comment. We agree that the lower transposon insertion density in LB could influence interpretation, though likely not to the extent of explaining the observed LB-specific gene set. As previously mentioned, we repeated the experiments until achieving sufficient insertion coverage for LB to meet our minimal quality criterion of at least 200,000 insertion sites, as determined during protocol optimization. If low coverage was the predominant factor behind LB-specific genes, they would predominantly be very small. The graph below represents the size of LB-specific *Hi-Fi* genes (Y-axis = number of genes, X-axis = size of genes). While genes smaller than the LB bin size are present and should be interpreted with caution (orange bar), they represent only a minority. Most LB-specific genes are of standard size, supporting their biological relevance rather than being artifacts.

- We initially considered performing a temporal comparison in LB; however, the reduced insertion density persists across all passages, making it difficult to conduct a robust time-resolved analysis similar to what we achieved in defined media. While the data at the final passage are sufficient for bin-based analysis, they are not adequate to confidently assess trends over time. For this reason, we focused our temporal analysis on the defined media, where coverage was consistently higher.

line 344 - It is not clear which genes are being referred to here by 'entire' - is this the median gene size for the "entire important genes" and "entire essential genes"?

- The sentence was meant to refer to every gene in *E. coli* and has been updated for clarity.

line 356-358 1551 bp is not a small gene?

- The classification of a “small” gene depends on the context. A gene of 1551 bp is actually in the top ~15% of the longest genes in the *E. coli* genome. The intent of the sentence was to emphasize that this gene is much larger than the shorter genes that typically account for discrepancies in Tn-seq experiments. The text has been adjusted to include this information.

Line 517 - Author D.M is not listed?

- Author D.M. has been moved from the author contributions statement to the acknowledgment with his consent, as his involvement was deemed more appropriate for acknowledgment rather than authorship.

Document only gives references up to no. 37. Beyond this could not be reviewed.

- An error in the citation manager software hid the latter half of the citations. The issue has been resolved in the current manuscript.

Reviewer #2:

Summary:

Champie et al. propose a new way to determine gene essentiality using a traditional transposon-seq approach in *E. coli*. They compare 3 different types of media; two are defined (MOPS and EZR) and one undefined (LB). Using this approach they define sets of genes that are essential in all three media conditions as well as genes that are specific to one or two of the media. Because they serially passage the transposon library through 5 rounds of growth (through mid-log and stationary phase), they are able to select against genes that are essential at any point during growth.

Based on the number of transposon insertions, they can classify genes as essential, somewhat essential, and non-essential. Instead of just looking at the total number of insertions per gene, they use a sliding window approach that is less sensitive to gene length. Employing the use of a third class of genes (i.e., the somewhat essential genes that impact overall fitness) helps to create a more nuanced view of the role of genes during growth.

We wish to thank Reviewer #2 for the attentive assessment of our work. The comments offered were valuable and inspired numerous clarifications and refinements across the manuscript. We trust that the revisions made in response to these points will satisfactorily resolve the issues raised and further enhance the manuscript's overall clarity and coherence.

Comments:

Line 35: Suggest “possible” instead of “accessible”.

- Suggestion accepted.

Line 42: Suggest “less of an” instead of “a lighter”. As a general note throughout, there are several more colloquial, subjective adjectives (i.e., “lighter”, “precarious”) used. It would be best to remove these and/or make sure that they are quantifiable.

- We acknowledge that terms such as “lighter” or “precarious” can sound subjective. Our intent was to convey the qualitative impact of gene inactivation and the broad categorization of genes according to their relative contribution to fitness. To improve clarity, we have replaced these adjectives with more precise comparative terms (high, low and none).

Line 118: I disagree with introducing the terminology of “important” over something more traditional like “fitness” genes, which should still capture the need for the genes at different growth phases. Although the screen done here looks at three in vitro media conditions, it could be argued that virulence genes may be quite important during infection, but non-essential in laboratory culture. As the *E. coli* strain used is a laboratory strain and most of the genes that had insertions tended to involve central carbon metabolism, “fitness” still makes more sense to me.

- Thanks for this comment. In line with the previous answer, we also revised the overall nomenclature of the categories to replace the notion of “importance for growth” with “fitness,” resulting in numerous changes throughout the text and figures.

Line 81: “providing a unique perspective on the question” is awkwardly worded. Do you mean that each strategy has different advantages and pitfalls. If so, it may also help to list some of these and how/why your strategy ameliorates some of the weaknesses or builds on the strengths.

- The aim of this paragraph was to highlight that despite the abundance of data generated about “essentiality” in *E. coli*, the various ways in which both the genes have been inactivated and the mutants selected is so vast that it is hard to compare existing datasets to find conditional important genes. The text has been adjusted and reorganized to support this point more clearly (Introduction, paragraph #3).

Line 128: Please include the enzyme name here and not just in the methods.

- Suggestion accepted, and a short description of the procedure was added for convenience.

Line 130: Please define “acceptable level”. What was the fold-change reduction?

- Following these great suggestions, we added a few sentences containing additional details to clarify the plasmid depletion and the improvement brought by this depletion. (Results, paragraph #3)

Line 141: It was unclear when the cultures were actually passed each time. Was stationary phase defined as overnight? A specific number of hours? An OD reading?

This is especially important to note given that three different media are used. I didn’t see this information in the methods either.

- Each passage lasts 24h to let each population reach a late stationary phase. All media have been passed at the same time. Thanks for the suggestion, we have added this information in the Results section, paragraph #1. Links for the complete protocols published on protocol.io are also included in the methods (Transposon mutagenesis and library preparation) section.

Line 172: How many Tn insertions do you estimate occur after 5 passages? How would you account for synthetic lethality phenotypes? This is touched on briefly in the discussion, but would like to see an estimate of the number of insertions per strain after continuous growth for 5 passages.

- All transposon insertions occur before passage 0, as the transposons are delivered via a suicide plasmid that activates upon entry into the recipient cells. Each plasmid can only activate once as the transposon leaves the plasmid upon activation. No new insertions are expected during subsequent passages as the conjugative suicide-plasmid delivery strain is unable to survive because of its diaminopimelic (DAP) auxotrophy and the kanamycine selection. The relative “increasing” number of sequenced insertions over time that can be seen in **Table S1** is due to an exponential diminution of the mutagenesis plasmid associated reads (which are diluted during passages) and the higher sequencing depth used at passage 5 (P5), which was prioritized to maximize insertion diversity for the *iScore* (renamed *fiScore*) analysis.
- Regarding synthetic lethality, even if some cells were to receive two transposons during the initial mutagenesis, they would have to be both inserted in genes involved in a synthetic relationship to disappear from the population during passages. The likelihood of which is low enough that it will not significantly affect overall insertion density. Synthetic lethality is a key focus of our group, and we explore this aspect in depth in a dedicated follow-up study, which we hope will be of interest to the reviewer.

Line 183 (Fig 2C): For this and subpanels like it throughout the manuscript, what is the x-axis or the 4 categories that are represented?

- For this figure and the following panels, the black circles in the UpSet chart indicate the sets of genes whose inactivation resulted in a negative fitness effect. In Figure 2C specifically, each vertical column represents an intersection between the two analytical approaches used to evaluate our EZR dataset. The black circles within a column show which method(s) identified a given set of genes as negatively impacting fitness, while the bar above each column corresponds to the number of genes in that intersection. Thus, the figure illustrates both the total number of negatively impacting genes identified by each approach and the degree of concordance or discordance between the two methods. Description of figure 2C has been modified for clarity.

Line 187: Why do you think this is true? I would imagine that there is less variation in the production and lots of salts and sugars compared to how yeast extracts are manufactured. Are there alternative explanations?

- Thanks for asking! Upon rereading, we realized the original statement was inaccurate. Our intention was to convey that the outcome of such a simulation would be inaccurate, rather than error-prone. This is because an accurate determination of the complete list and concentration of individual molecules in LB medium is currently unavailable, and experimentally difficult to obtain. As yeast extract is roughly dried up yeast cytoplasm it contains the numerous molecules required for yeast metabolism, the exact list of which is not available for each batch. While approximations of LB composition can be made, any predictions generated by a metabolic model are inherently limited by the accuracy of the simulated medium's composition. We have clarified this sentence (Definition of a Fitness Impact metric, paragraph #3).

Line 195: What was the number of previously known genes rather than “most”? Is there a succinct way to describe what number and types of genes are meant by “additional genes”?

- Good points. We have added specific numbers to the text and to avoid overcrowding, we refer readers to Figure 3C for detailed information. Preliminary analysis using Panther 19.0 indicates that these additional genes are enriched in ATP biosynthetic processes and aerobic respiration. They represent a diverse set of genes that are not strictly essential but whose disruption leads to a measurable fitness decrease, for example, genes in the *nuo* operon. We thank the reviewer for this comment. The observed enrichment in these pathways further supports our argument that gene function and fitness impact can vary depending on the specific medium conditions. Those modification were included in the *Definition of a Fitness Impact metric* section, paragraph #3.

Line 197: Please include a short descriptor of what the *nuo* operon and *nuoF* encode. Would you expect a mostly amino acid/peptide media like LB to require this moreso than the more carbohydrate heavy media?

- We have added a description of the *nuo* operon and its role in aerobic respiration (*Definition of a Fitness Impact metric* section, paragraph #3). Given that this operon is part of the electron transport chain, a core component of aerobic metabolism, we initially expected that its inactivation would impair fitness similarly across all three tested media. Surprisingly, however, our data revealed a pronounced fitness defect for $\Delta nuoF$ mutants exclusively in LB medium. This unexpected medium specificity in such a fundamental metabolic pathway prompted us to highlight the *nuo* operon as a striking example. We have added traces of this reasoning in the text in the same paragraph (*Definition of a Fitness Impact metric* section, paragraph #3).

Line 216: Here and throughout (although this gets better in the second half of the manuscript), please give examples of genes—especially if it is a manageable number like 2.

- Following this good suggestion, we have added a brief sentence to provide context for the core poorly characterized genes. In addition, we have made various changes throughout the manuscript to better contextualize the functions of the genes within each group, enhancing the clarity and interpretability of our findings.

Line 218: Define “accurate”. How do you know what the ground truth is?

- While we do not have access to a definitive ground truth, we observed that several genes such as *cvpA*, *bax*, or *elaA* to name a few, are categorized as completely uncharacterized in the KEGG database, yet have partial or full functional annotations in EcoCyc. Each database offers distinct strengths that we leveraged accordingly: KEGG provides a streamlined and structured gene ontology system, which we used to categorize gene functions for medium specific gene groups. In contrast, EcoCyc has more detailed gene-level information, thanks to its regular manual curation of literature. To avoid confusion, we have modified the text to reflect that we now only use KEGG for ontological analyses. All characterization level estimation were only sourced from EcoCyc.

Line 225: Instead of “color matching” please just list out what the colors correspond to. It took a while to understand that you mixed the two colors assigned to the media. Great way to help remember it when looking at the figure, but hard to interpret as worded in the legend.

- Great comment, the legend of the figure has been clarified.

Line 232: Please define where (i.e., what media conditions) the numerator and denominator represent in the legend.

- This panel is dedicated to the core important gene, i.e. genes important in all 3 conditions. The legend of the figure has been clarified following this comment.

Figure 6: Here, and in supplemental, what is the difference between the neutral/gray categories?

- In the context of finding the list of genes required to maintain efficient growth in specific media, there is no functional difference among the three (now four after deciding to remove the Precarious group based on a previous comment) neutral (now *No-FI*) categories. Because the labeling was based on five clusters (see comment below and Fig. S6A), we ended up with four groups under our *No-FI* definition (characterized by relatively high insertion levels and/or positive (or almost) temporal variation rate), which we kept differentially colored for transparency.
- It is possible that biological differences exist among genes in these four *No-FI* clusters, as they exhibit varying depletion rates and insertion levels. One factor driving this subdivision might be genes that confer different degrees of fitness to increase when inactivated. We initially considered segregating these into separate groups, but our method proved surprisingly limited in isolating cases that are plain to see when manually inspected. Since the primary goal of this study was to define medium-specific minimal gene sets, we chose not to risk an approximate fitness increase classification. Other approaches to granularly classify such subtle cases using read counts are under development but were not ready to be included in this manuscript.

Lines 302-304: Where do most of the insertions occur? If most are in the beginning of the gene, you might expect to see a lethal phenotype, but potentially if they are mostly toward the end and the *E. coli* is growing in a rich media, the abundance of lysine may be able to compensate for slowed charging, whereas it may not be able to in minimal media and/or if it were relying on making its own lysine from scratch.

- As can be seen in the screenshot below, *lysU* (top image) exhibits a high density of transposon insertions across all three media conditions. This suggests that insertions are tolerated throughout the gene, including potentially early regions. In contrast, *lysS* (bottom image) shows an overall absence of insertions in EZR and MOPS, with insertions in LB. In both images y-axis represents the number of reads per insertion and is constant across all tracks (linear scale cropped at 10 reads).

- The differential fitness impact observed between LB and EZR is indeed intriguing, especially since both media supply lysine. One potential explanation could be that LB, which contains yeast extract, may provide precharged tRNA^{Lys} or other components that buffer the effects of impaired tRNA charging. This could mitigate the fitness cost of transposon insertions in *lysS*. In contrast, EZR, while supplemented with lysine, may lack such compensatory factors, leading to a more pronounced fitness defect. Such specific cases are the subject of a follow-up article which we hope will be of interest to the reviewer.

Line 325: Suggest changing “coherent” with “consistant”.

- The text has been corrected as suggested, thanks.

Line 357-358: Please reword or delete the awkward phrasing at the end of the sentence to improve clarity.

- Thanks for pointing it out, this sentence has been removed from the text as its content is implicit in the previous sentence.

Line 399: Although it is described elsewhere and you link to protocols.io, please include a brief summary of what transposon was used and how the mutagenesis was performed.

- A short description of the protocol and the transposon have been added in the methods (*Transposon mutagenesis and library preparation* section).

Line 415: Please include the GenBank/ENA accession number used.

- Good point, NCBI accession number of the reference genome has been added to the text.

Line 478: Some words are missing: “was lower than”.

- Thanks, the sentence was rewritten as it was convoluted and prone to misinterpretation.

Line 480: Please include a brief description of what is in the reference.

- A brief description of reference #58 has been added.

Line 496: How were 5 clusters decided upon? Statistically? Some biological reason?

- Based on this great question, we realized that the dendrogram supporting the choice of the five clusters was missing, it was therefore added in Fig S6A.

Sup. Fig 1: Since this data is based off of 20 individual cultures per medium, suggest showing number of mutations per culture as dots and adding a standard deviation.

- Thanks for this great suggestion, we have accordingly updated the figure S1 to show the distribution of insertion sites found in each replicate instead of the total of insertion for the passage. Although the number of insertions seems to vary, we shown in an earlier publication that the inter-sample variation between replicates produced by the HTTM method is highly consistent (<https://doi.org/10.1371/journal.pone.0283990>).

Sup. Fig 2: Suggest changing the box that says “Table_S2” to something more descriptive, since this is a schematic.

- Thanks, the caption has been updated to Medium-specific genes modules.

Re: mSystems01425-25R1 (Comparative Analysis of Gene Importance in Escherichia coli Across Growth Conditions)

Dear Prof. Sébastien Rodrigue:

Thank you for sending your revised manuscript. One reviewer has identified a small number of clarifications in the methods and results sections that are important to address prior to acceptance.

Revision Guidelines

Sincerely,
Julia Willett
Editor
mSystems

Reviewer #1 (Comments for the Author):

Thank you for clarifying a couple of points, and doing so with so much detail.

Reviewer #2 (Comments for the Author):

Summary:

Champie et al. developed an algorithm to identify the core essential genes in *Escherichia coli*. They used a combination of transposon mutagenesis and growth in 3 different types of media: LB, MOPS, and EZR, which contains additional defined components in a MOPS media base. Growth in different media selected for 412 core essential genes in all three tested media; however, some genes (or gene termini) were essential in only 1 or 2 of the tested media. Instead of calculating essentiality based on total number of transposon insertions per gene, the authors used a sliding window approach to chunk the genes into segments. This allowed them to determine areas of the genome that are particularly sensitive to transposon insertion at higher (sub-gene) resolution and mitigate limitations of read depth or few insertions. This allows the authors to identify genes that are very susceptible to insertions (i.e., HITS) and those that are not (i.e., MISSs). Determining the core set of genes needed in different media will help in designing the minimal components needed for bioengineering strains.

The authors addressed the previous reviewers queries well. However, some minor comments remain.

Comments:

Line 138: Verb tense typo: "are only 16 AvrII sites"

Line 162: Several typos.

Figures: Assuming that "Nb" is an abbreviation for number. Would be clearer to use the more common "No." instead. Otherwise this abbreviation should be stated in the relevant legends.

Lines 217-219: Suggest adding some specific limitations here. For instance, in rich media where other, "WT" cells are growing quickly, not being able to use aerobic respiration may be more of a disability compared to in minimal media where the "WT" cells are growing slower.

Line 243: Is "Representation" meant instead of "repartition"?

Line 263: typos: "sections in the 3' end of these genes"

Line 269-270: "...they are considered HI-FI genes in the final..."

Line 291: Please include a definition of why you would be "confident". Is there a particular threshold score that is used?

Line 307: "visually indistinguishable" can be removed.

Lines 344-348: Is lysU transcribed at the same level of lysS? How does this relate to codon usage or potentially the types of genes/codons that may favor one synthetase over the other? Highly unlikely that aminoacylated tRNAs in the media could be imported and used in the cell.

Line 369: "assumed to be" can be removed.

Line 398-399: Do you have an idea for a threshold that would define how impaired a strain would have to be before it was detected?

Lines 441-449: Methods should be written in the past tense.

Line 449: Was the enzymatic or fragmentation protocol used for the NEB kit? What was the input DNA amount and how many PCR cycles were used?

Line 450: Why was Nextera used if the NEB kit was used? Do you mean the NextSeq platform?

Line 469: Please include read size.

Figure S5: Is there evidence of internal promoters or sRNAs that may explain why some gene termini seem to be essential in different media?

Summary:

Champie *et al.* developed an algorithm to identify the core essential genes in *Escherichia coli*. They used a combination of transposon mutagenesis and growth in 3 different types of media: LB, MOPS, and EZR, which contains additional defined components in a MOPS media base. Growth in different media selected for 412 core essential genes in all three tested media; however, some genes (or gene termini) were essential in only 1 or 2 of the tested media. Instead of calculating essentiality based on total number of transposon insertions per gene, the authors used a sliding window approach to chunk the genes into segments. This allowed them to determine areas of the genome that are particularly sensitive to transposon insertion at higher (sub-gene) resolution and mitigate limitations of read depth or few insertions. This allows the authors to identify genes that are very susceptible to insertions (i.e., HITs) and those that are not (i.e., MISSs). Determining the core set of genes needed in different media will help in designing the minimal components needed for bioengineering strains.

The authors addressed the previous reviewers queries well. However, some minor comments remain.

Comments:

Line 138: Verb tense typo: “are only 16 AvrII sites”

Line 162: Several typos.

Figures: Assuming that “Nb” is an abbreviation for number. Would be clearer to use the more common “No.” instead. Otherwise this abbreviation should be stated in the relevant legends.

Lines 217-219: Suggest adding some specific limitations here. For instance, in rich media where other, “WT” cells are growing quickly, not being able to use aerobic respiration may be more of a disability compared to in minimal media where the “WT” cells are growing slower.

Line 243: Is “Representation” meant instead of “repartition”?

Line 263: typos: “sections in the 3’ end of these genes”

Line 269-270: “...they are considered HI-FI genes in the final...”

Line 291: Please include a definition of why you would be “confident”. Is there a particular threshold score that is used?

Line 307: “visually indistinguishable” can be removed.

Lines 344-348: Is *lysU* transcribed at the same level of *lysS*? How does this relate to codon usage or potentially the types of genes/codons that may favor one synthetase over the other? Highly unlikely that aminoacylated tRNAs in the media could be imported and used in the cell.

Line 369: “assumed to be” can be removed.

Line 398-399: Do you have an idea for a threshold that would define how impaired a strain would have to be before it was detected?

Lines 441-449: Methods should be written in the past tense.

Line 449: Was the enzymatic or fragmentation protocol used for the NEB kit? What was the input DNA amount and how many PCR cycles were used?

Line 450: Why was Nextera used if the NEB kit was used? Do you mean the NextSeq platform?

Line 469: Please include read size.

Figure S5: Is there evidence of internal promoters or sRNAs that may explain why some gene termini seem to be essential in different media?

Response to reviewers' comments

We would like to once again thank the reviewers for their thorough and constructive feedback. We greatly appreciate their contribution to strengthening the scientific quality of our manuscript. Below, we provide detailed responses to each point raised in this second round of review. For ease of reading, reviewers' comments are presented in blue, with our responses immediately following in black.

Reviewer #2

Comments:

- Line 138: Verb tense typo: "are only 16 Avrll sites"
 - Verb tense was adjusted.
- Line 162: Several typos.
 - The typos in the legend of Figure 1 have been corrected.
- Figures: Assuming that "Nb" is an abbreviation for number. Would be clearer to use the more common "No." instead. Otherwise this abbreviation should be stated in the relevant legends.
 - We have added the definition of Nb in the legend of the appropriate figures and will use No. in future manuscripts.
- Lines 217-219: Suggest adding some specific limitations here. For instance, in rich media where other, "WT" cells are growing quickly, not being able to use aerobic respiration may be more of a disability compared to in minimal media where the "WT" cells are growing slower.
 - This could indeed be one possible explanation for the observed medium-specific fitness impact of the *nuo* operon. However, we would prefer to avoid providing an overly specific interpretation, as numerous factors beyond differential growth rates may contribute to this specificity. It is also worth noting that growth in EZR is only slightly slower than in LB, which limits how much the difference in growth rate alone can account for the effect.
- Line 243: Is "Representation" meant instead of "repartition"?
 - We replaced repartition by Representation.
- Line 263: typos: "sections in the 3' end of these genes"
 - The missing word has been added
- Line 269-270: "...they are considered HI-FI genes in the final..."
 - The phrasing has been adjusted
- Line 291: Please include a definition of why you would be "confident". Is there a particular threshold score that is used?
 - We used the term "confident" because, given the sensitivity of our approach, any gene with a measurable fitness impact would be expected to fall within the previously defined Hi-FI or Lo-FI categories. We have reworded this sentence and clarified that the assignment to the No-FI group simply reflects the absence of evidence placing a gene into one of the other FI categories.

- Line 307: "visually indistinguishable" can be removed.
 - The words were removed.
- Lines 344-348: Is *lysU* transcribed at the same level of *lysS*? How does this relate to codon usage or potentially the types of genes/codons that may favor one synthetase over the other? Highly unlikely that aminoacylated tRNAs in the media could be imported and used in the cell.
 - A more in-depth review of the literature confirmed that although both *lysS* and *lysU* are individually capable of supporting growth, *lysU* expression in wild-type strains is extremely low and insufficient to compensate when *lysS* is absent. We have updated the manuscript accordingly to provide a more accurate description of *lysU* regulation.
 - Regarding the reduced fitness impact observed in LB, our literature analysis did not reveal a definitive mechanistic explanation. *lysU* is known to be subject to several regulatory inputs (including induction at elevated temperatures, under anaerobic conditions, and at low pH), and it remains unclear whether any of these activation conditions, or any other yet undiscovered one, might be specifically triggered in LB medium. In the absence of conclusive evidence, we refrain from overinterpreting this aspect and instead present the updated information on *lysU* regulation more cautiously.
- Line 369: "assumed to be" can be removed.
 - We would prefer to keep the wording as it is. Given that we don't have the real composition of LB, it is difficult to be fully assertive about the "completeness" of LB vs other media.
- Line 398-399: Do you have an idea for a threshold that would define how impaired a strain would have to be before it was detected?
 - We cannot define an exact universal threshold for detectability because several factors, most notably gene size, might affect how our method classifies genes. Nonetheless, our validation allows us to approximate the lower limit of what can be reliably detected. For instance, fitness defects of ~6% (such as the growth reduction observed for *pdeB*) lie close to the minimal effect our approach can resolve. In our validation experiments, strains exhibiting a similar magnitude of impairment, such as *yebG* in MOPS, were successfully detected. Importantly, we did not miss any genes with fitness defects larger than this. Based on these observations, we could estimate that our practical detection limit is approximately 6% growth impairment.
- Lines 441-449: Methods should be written in the past tense.
 - Methods were corrected with the appropriate tense.
- Line 449: Was the enzymatic or fragmentation protocol used for the NEB kit? What was the input DNA amount and how many PCR cycles were used?
 - The fragmentation was performed using the Ultra II FS enzymatic fragmentation mix from the NEBNext Ultra II kit. Each library was prepared with approximately 80 ng of genomic DNA as input. If the reviewer is referring to the *AvrII* digestion, this enzymatic treatment is performed after adapter ligation as an additional step specific to our HTTM workflow. After ligation, the remaining steps do not follow the manufacturer's protocol; the two PCR steps used in our workflow consist of 20 cycles and 5 cycles, respectively. These procedural details were not included in the manuscript as the full protocol has already been published, and the complete step-by-step version is available on Protocols.io as referenced in the Materials and Methods section (see line 469 of the revised manuscript).
- Line 450: Why was Nextera used if the NEB kit was used? Do you mean the NextSeq platform?

- We used the NEBNext Ultra II kit for DNA fragmentation and adapter ligation, but the sequencing itself relies on Illumina Nextera-type sequencing primers.
- In our HTTM system, the Nextera read structure is crucial because Nextera sequencing primers initiate reading directly from the first base adjacent to the Tn5 mosaic end sequence located at the extremity of the transposon. Since this transposon end is embedded within our construct, using Nextera-compatible primers ensures that sequencing begins immediately at the genomic junction. The manuscript describing the high-throughput transposon mutagenesis (HTTM) protocol that we previously developed and now use in this manuscript, which is cited at the beginning of this section, includes a visual illustration of what happens to DNA fragments during library preparation.
- The libraries were sequenced on an Element AVITI instrument, which is compatible with Illumina Nextera sequencing primers (Cloudbreak Freestyle). This has now been clarified in the Methods section.

- **Line 469: Please include read size.**
 - Read size has been included.

- **Figure S5: Is there evidence of internal promoters or sRNAs that may explain why some gene termini seem to be essential in different media?**
 - For the specific genes shown in Figure S5, we consider it unlikely that the differences observed in insertion density near gene termini reflect internal promoters or sRNA activity. Those genes are convergent, and we did not identify any annotated promoters in the relevant regions using available databases. The overall transposon insertion signal in these termini is relatively low, making it more consistent with inter-medium variation or experimental noise rather than a biologically meaningful pattern.
 - That said, we agree with the reviewer that genuine cases of essential sub-regions within genes may indeed arise from regulatory elements of neighboring genes, such as internal promoters, untranslated regions, or embedded sRNAs. This is an important possibility that warrants deeper investigation, and we plan to explore it in future work.

Re: mSystems01425-25R2 (Comparative Analysis of Gene Importance in Escherichia coli Across Growth Conditions)

Dear Prof. Sébastien Rodrigue:

Your manuscript has been accepted, and I am forwarding it to the ASM production staff for publication. Your paper will first be checked to make sure all elements meet the technical requirements. ASM staff will contact you if anything needs to be revised before copyediting and production can begin. Otherwise, you will be notified when your proofs are ready to be viewed.

Sincerely,
Julia Willett
Editor
mSystems